# Harnessing the benefits of diversity to address socio-environmental governance challenges

Jacopo A. Baggio[1,2]☯*, Jacob Freeman[3,4]☯, Thomas R. Coyle[5], John M. Anderies[6,7]

**1** School of Politics, Security, and International Affairs, University of Central Florida, Orlando, Florida, United States of America, **2** National Center for Integrated Coastal Research, University of Central Florida, Orlando, Florida, United States of America, **3** Anthropology Program, Utah State University, Logan, UT, United States of America, **4** The Ecology Center, Utah State University, Logan, UT, United States of America, **5** Department of Psychology, University of Texas at San Antonio, San Antonio, TX, United States of America, **6** School of Sustainability, Arizona State University, Tempe, AZ, United States of America, **7** School of Human Evolution and Social Change, Arizona State University, Tempe, AZ, United States of America

☯ These authors contributed equally to this work.
* jacopo.baggio@ucf.edu

**Data Availability Statement:** Data, model and analysis code can be found at https://github.com/jb80/CpxCogDiv and as supplementary files.

**Funding:** JAB, JF and TRC acknowledge support from the National Science Foundation Grant SMA-

## Abstract

Solving complex problems, from biodiversity conservation to reducing inequality, requires large scale collective action among diverse stakeholders to achieve a common goal. Research relevant to meeting this challenge must model the interaction of stakeholders with diverse cognitive capabilities and the complexity of the problem faced by stakeholders to predict the success of collective action in various contexts. Here, we build a model from first principles of cognitive abilities, diversity, and socio-environmental complexity to identify the sets of conditions under which groups most effectively engage in collective action to solve governance problems. We then fit the model to small groups, U.S. states, and countries. Our model illustrates the fundamental importance of understanding the interaction between cognitive abilities, diversity, and the complexity of socio-environmental challenges faced by stakeholders today. Our results shed light on the ability of groups to solve complex problems and open new avenues of research into the interrelationship between cognition, institutions, and the environments in which they co-evolve.

## Introduction

In a globalized, interdependent, and fast changing world, *diverse stakeholders* must work in concert to govern resources and avoid crossing local and/or planetary boundaries [1–4]. Managing resources requires coordination, and, often, cooperation among multiple stakeholders, such as: businesses, non-governmental organizations, and governments. Diverse stakeholder groups bring to the table different technologies and institutions, background experiences, values, and levels and types of cognitive abilities leading to, at times, divergent cognitive representations of a system. While the benefits (e.g., increased system understanding, cooperation, ability to find novel solutions) and costs (e.g., conflict, stalled positions) of diverse stakeholders

1620457. https://www.nsf.gov/awardsearch/showAward?AWD_ID=1620457&HistoricalAwards=false The funders had no role in study design, data collection and analysis, decision to publish, or preparation of the manuscript.

**Competing interests:** The authors have declared that no competing interests exist.

cognitive tools and abilities have been studied by psychologists, organizational scientists, economists, and sustainability scientists [5–16], to the best of our knowledge, formal models of how the multiple dimensions of stakeholder diversity (technologies, background experiences, values, and cognitive abilities) interact to affect collective problem solving in a given socio-environmental context are lacking. Here, we help fill this knowledge gap by first formally modeling the joint effects of socio-environmental complexity and three dimensions of stakeholder diversity that we call cognitive abilities, cognitive representational diversity, and cognitive tools on the ability of groups to find collective solutions to governance problems. Then, we partially evaluate the plausibility of the model by fitting it to data from small groups, U.S. states, and countries. Our model and model fitting exercise make a significant contribution to the understanding of collective action and the ability of groups to solve problems by synthesizing results from psychology, organizational science, economics, and sustainability science into a coherent, formal model, grounded by the Institutional Analysis and Development Framework (IAD).

The IAD framework organizes the study of collective action (Fig 1) [17, 18]. Collective action refers to the capability of groups–at a relevant scale–to work together to solve a problem or a range of ever shifting problems. Within the IAD, collective action takes place within an Action Situation. This Action Situation defines the social space in which individuals interact,

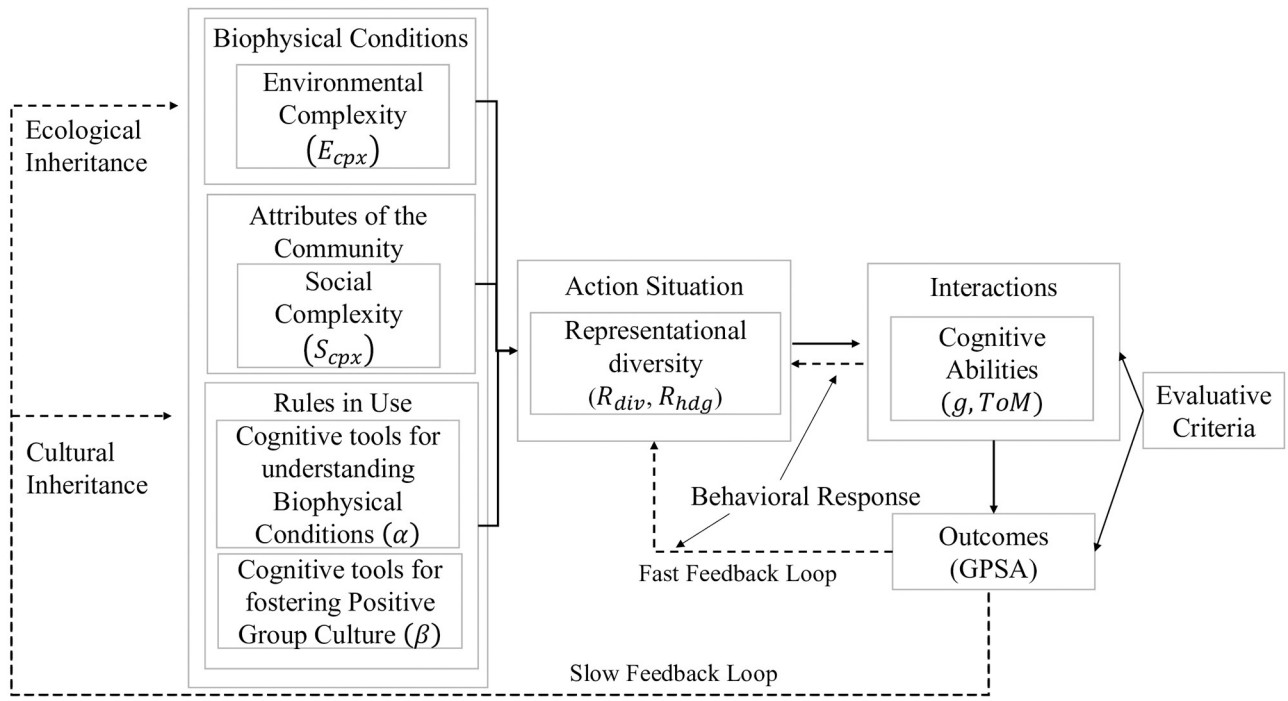

**Fig 1. The IAD framework (adapted from [17]) depicts how slow variables (i.e. biophysical conditions, attributes of the community, and rules in use) generate action situations, which, in turn, generate interactions among agents for collective decision making and problem solving.** In the adapted framework, we specify environmental complexity $E_{cpx}$ as being a key attribute of biophysical conditions, social complexity $S_{cpx}$ as a key attribute of the community, and cognitive tools as a factor affecting rules in use. Cognitive tools refer to the mean traits of a group inherited as a consequence of past technological adaptations and social interactions. We define two sets of tools: (1) traits that can foster a better understanding of the biophysical environment and its dynamics ($\alpha$), and (2)traits that foster a positive group culture ($\beta$). Complexity and cognitive tools result from cultural inheritance and constitute parameters that impact interactions between and within groups in specific action situations. Interactions in action situations depend on specific cognitive abilities (here the ability to correctly represent key dynamics within the environment ($g$), and the ability to model others intentions ($ToM$). Interactions, together with complexity and cognitive tools affect the representational diversity ($R_{div}$) as well as the ability of groups to harness such diversity ($R_{hdg}$) in an action situation, and give rise to specific outcomes that are partly ascribed to a group's problem solving ability ($GPSA$). Outcomes are then evaluated via specific criteria and potentially modified via learning and collaboration in the behavioral (fast) feedback loop.

either positively or negatively, to generate governance outcomes. Outcomes generated by the success or failure of collective action are then assessed by specific Evaluative Criteria (e.g., was enough public good provided) and corrective measures may be put in place.

Outcomes generated by Action Situations, the center of the IAD, result from two key feedback loops: (1) a fast behavioral response loop, and (2) a slow cultural and ecological inheritance loop (see Fig 1). The behavioral response loop refers to how groups adjust their actions to their given circumstances over days to a few years. Individuals interact, reach specific outcomes, evaluate, and respond. Community responses to epidemics can be thought of an example of the behavioral response loop. Individuals within some defined community, say a school district, change (or not) their behavior based on how they interpret the system (in this case how the epidemic is spreading) and their assessment of other's intentions to cooperate (wear masks, get vaccinated, etc.). The adjustments and evaluations that individuals make, we contend, draw on the cognitive tools and abilities that they bring with them into an Action Situation. The cultural and ecological inheritance loop is defined by slower feedback processes in which behavioral adjustments affect the cultural dimensions of rules in use (e.g., institutions, accepted technologies) and attributes of a community (e.g., diversity of experiences, values, beliefs and objectives), as well as the biophysical dimensions of a system (e.g., changes in sealevel or snowpack).

Over decades to millennia, cultural inheritance influences social complexity and ecological inheritance affects environmental complexity. Treating these slow processes as parameters, it follows that the complexity of an Action Situation faced by a group varies along two dimensions: (1) The number of social units involved with different perspectives, knowledge, values, beliefs, and objectives affecting the overall social complexity ($S_{cpx}$) of an Action Situation. (2) The number of biophysical processes relevant to the problem that a group must solve, and the interaction of these biophysical processes (linear, non-linear, presence of uncertainty, and thresholds) affecting the overall environmental complexity ($E_{cpx}$) of an Action Situation. As the overall complexity of an Action Situation increases, collective action and problem solving become more difficult. This is because the more complex a system, the more uncertainty individuals and groups have about how to plan and act. According to research in psychology, organizational science, and economics, groups can draw upon multiple resources to learn (i.e., adjust behavior over the short-term) and, potentially, generate better collective problem solving outcomes [5, 10, 19]. These resources include what we call cognitive abilities, cognitive representational diversity, and cognitive tools developed over time in the social-ecological contexts that generate repeated, structurally similar Action Situations. Understanding the differences and relationships between these three concepts lies at the core of the model that we propose.

Cognitive abilities refer to variation in the capacities of individuals to construct mental models of biophysical and social environments. Two of the most general abilities, discussed below, are general and social intelligence, theory of mind in particular. Cognitive representational diversity refers to the different ways that individuals describe and understand a system in terms of physical mechanics and values or legitimate objectives. For instance, two individuals may have a different understanding of water variability within an irrigation system, how the system may react to such variability, and values associated with water allocation. One individual may see the system as robust to predicted changes and value efficiency in water allocation and withdrawal. Another individual may assess the system as insufficient to deal with predicted changes in water availability and value longer term investments in the physical infrastructure that may sacrifice short-term water extraction to gain higher water extraction in the future. Finally, cognitive tools refer to the ability of groups to enhance cognitive abilities through technology, education, and institutions that help encode and monitor the dynamics of

biophysical systems or foster a positive group culture, given a past history of trust and shared narratives. Groups draw on these three resources to solve the biophysical and social challenges that arise in an Action Situation.

## Solving biophysical challenges

To model and solve biophysical challenges in an Action Situation, groups may draw on their general intelligence ($g$)–sets of correlated mental abilities, including spatial reasoning, impulse control, and the rational weighting of costs and benefits of options [20]–to learn and reason about a gravity driven irrigation system or the logic of a prisoner's dilemma [21]. These abilities are moderated by cognitive tools; such as education in numeracy or technologies that control the flow of water and make floods more predictable that we capture below with the parameter $\alpha$. The relevant tools or hardware will vary from society-to-society. Holding such tools equal, greater $g$ at the level of the group (often measured as the mean of all individual level general intelligence values) leads to a greater understanding of a system [11, 13, 21], thus less specific knowledge sets and more general, overlapping knowledge sets.

Groups may also draw on their diverse ways of representing a system (i.e., different background experiences, modes of inquiry, and spatial vs. verbal system descriptions) to form a better understanding of an ecological system's dynamics [6, 7, 10]. This mechanical dimension of cognitive representational diversity supercharges learning and leads to better representations of a system for decision making [5, 9, 10]. In fact, groups formed by individuals who adopt different perspectives approach problems differently and, if they are able to communicate and form shared values/objectives (see below), such groups are, on average, more apt to find solutions to problems than groups with similar levels of ability but lower diversity [22]. In sum, provided individuals share values and objectives, groups with high g and high cognitive representational diversity should best explore a system's problem space reducing the complexity of an environment by developing functionally relevant models of a system for solving governance problems.

Similarly, the need for cognitive representational diversity ($R_{div}$) goes beyond its effect on reducing environmental complexity. As the social complexity of an Action Situation increases (i.e., the number of social units involved with different knowledge, background experiences, and values/objectives grows), so does the need for including stakeholders who may have different values and objectives in decision making processes. This is because solutions to problems in Action Situations should be perceived as legitimate and fair, and, hence, inclusion criteria of all relevant stakeholders is a necessary condition for exploring an Action Situation's problem space [23]. A powerful way to increase the potential of a group to learn is to increase a group's cognitive representational diversity of the mechanics of an Action Situation, allowing for diverse and functionally relevant representations of the system (i.e. system representations that are at least partially correct and that are not willingly misleading). Such mechanical representations form the foundation of causal models useful for predicting future outcomes and making governance decisions, as long as stakeholders with potentially different values and objectives are included in decision making. However, even if groups include stakeholders with the diverse cognitive representations necessary to supercharge learning and provide a process whereby individuals with different values can participate in decision making, a necessary condition for legitimacy, this is only half that battle.

## Solving social challenges

Increasing the cognitive representational diversity of a group, necessary to supercharge learning and legitimacy, can also mean increasing the probability of including groups/individuals

with potentially incompatible objectives and value systems. For example, the higher the number of different social units (e.g., farmers, businesses, environmental groups, government agents, etc. all acting in the same Action Situation), the higher the probability of diverse experiences, modes of inquiry, and representations of a system, but also, different objectives and value systems [24, 25]. While diverse representations can supercharge learning, they can also be the result of divergent goals and/or value systems that lead to representational gaps. Representational gaps are defined as inconsistencies between how individuals perceive and assess the objectives of a system, furthering, in some cases, stalled positions and negatively affecting the possibility of forming joint goals and achieving solutions [24, 25]. To blunt this effect, groups can draw on their social cognitive abilities, moderated by the cognitive tools of shared norms and institutions that generate higher levels of trust and common narratives ($\beta$), to form shared goals [13, 16, 26, 27].

For example, repeated results from common pool resource experiments illustrate that when groups communicate–which is essential for sharing goals–individuals better form joint goals and resources deplete less quickly [27]. At the same time, restricting the ability to monitor and observe the behavior of others can reduce collective action [27, 28]. Finally, groups with higher levels of theory of mind (*ToM*) display an increased ability to find collective solutions and solve tasks in small-groups [29–31], govern a common pool resource more effectively [11, 13], and display higher levels of cooperation across a range of contexts [32–35]. Theory of mind is the ability to couple with the mental states and anticipate the preferences of other actors in a system [36]. Highly developed theory of mind means richer models of the mental states of other actors, and is proposed to reduce conflict, increase communication effectiveness, and reduce the costs of forming joint goals [11, 13, 14, 37, 38].

In sum, groups face an inherent exploration–exploitation tradeoff driven by an Action Situation's socio-environmental complexity (which is itself an outcome of long-term, cultural and ecological inheritance). Greater socio-environmental complexity requires that groups increase the cognitive representational diversity in an Action Situation to supercharge learning and legitimacy to effectively explore a system's social-ecological dynamics. However, more cognitive representational diversity also generates a greater potential for divergent interests and difficulty finding effective governance solutions due to blocking and conflict. Culturally mediated general and social intelligence are critical cognitive resources that groups can draw upon to blunt this exploration-exploitation tradeoff and learn effective solutions to governance challenges. Below, we model the average capability of groups to solve a governance problem in the behavioral feedback loop and explore the benefits and limits of culturally mediated general and social intelligence for blunting the exploration-exploitation tradeoff inherent to more complex socio-environmental systems.

## The formal model

We formally model the ability of groups to solve problems in two steps. In step one, we start with the need for diverse cognitive representations of a biophysical problem in a given Action Situation (aided by *g*). In step two, we add the need for individuals in groups to generate joint goals and shared vision (aided by *ToM*).

**Representational diversity.** The need for representational diversity ($R_{div}$) to solve socio-environmental challenges (and more generally complex problems) depends upon the social complexity ($S_{cpx}$), environmental complexity ($E_{cpx}$), and the ability of individuals within groups to make sense of biophysical processes by drawing on their *g* ($w_{div}(g)$), where $w_{div}$ is the ability of a group to generate functionally relevant representations of the system; *g* is the average group *g* and *max*(*g*) is the theoretical maximum value that group level *g* can reach. Eqs (1)

and (2) capture this process, i.e.,

$$R_{div} = S_{cpx} + (E_{cpx}/w_{div}(g)) \tag{1}$$

$$w_{div}(g) = \max(g)\left[\tanh\left(\frac{g}{\max(g) - g}\right)\right]^{\alpha} \tag{2}$$

where $0 \leq R_{div} \leq (S_{cpx} + E_{cpx})$, $S_{cpx} \geq 0$, $E_{cpx} \geq 0$, $w_{div} \geq 1$, $\alpha \geq 0$, and $g \geq 0$.

Eq 1 states that the need for representational diversity to solve a collective governance problem increases as the social ($S_{cpx}$) and environmental complexity ($E_{cpx}$) of an Action Situation increases. However, the ability of individuals to make sense of environmental complexity is, as discussed above, directly related to general intelligence. Thus, Eq 1 weights the need for representational diversity to solve a governance problem by $w_{div}$, at a given level of environmental complexity.

Eq 2, depicted in Fig 2, states that the ability of groups to reduce environmental complexity through learning in the behavioral feedback loop increases as $g$ increases. This, in turn, means that the need for cognitive representational diversity defined in Eq 1 declines as $g$ increases. Note that the relationship between $g$ and $w_{div}$ is non–linear and is moderated by $\alpha$ affecting the $g \rightarrow E_{cpx}$ relationship. The parameter $\alpha$ represents the cognitive tools affecting the ability of groups to generate functionally relevant representations of a system, and, thus, increase their understanding of the biophysical conditions and dynamics, such as: level of education and technology. Here we assume that the higher the value of $\alpha$, the more restrictions are imposed on accessing such cognitive tools. These characteristics affect the minimum level of group $g$ needed, as well as the level of $g$ needed to reduce the uncertainty associated with

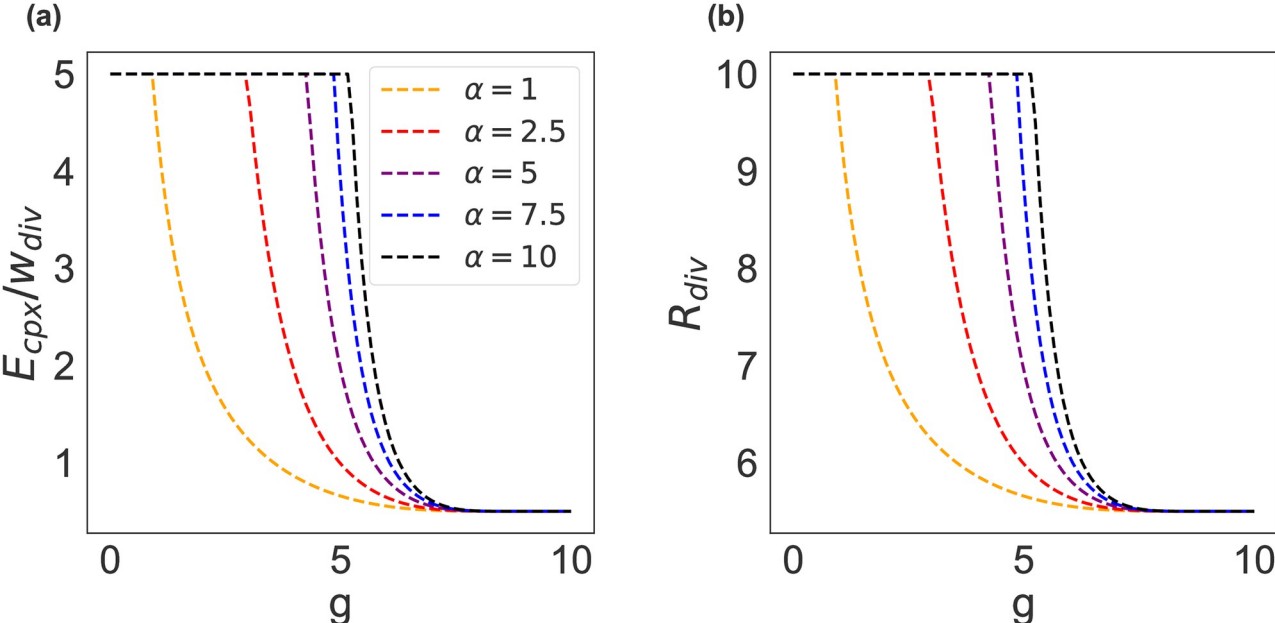

**Fig 2. Effect of group unobserved characteristics $\alpha$ and general intelligence ($g$) on the need for representational diversity ($R_{div}$).** Panel (a) shows how the 'environmental component', $E_{cpx}/w_{div}$ changes with $g$. A higher value of $\alpha$ indicates a higher incidence of restricted access to schooling and technology. Thus, much higher $g$ is needed for an individual to understand a system and reduce the need for representational diversity. Panel (b) shows the effect of $S_{cpx}$, shifting the curves, on the need for representational diversity ($R_{div}$) and, thus, the overall $R_{div}$ taking into account both environmental and social complexity. Figure was generated keeping $E_{cpx} = S_{cpx} = 5$.

environmental complexity. The effect of group characteristics, which are the outcome of long-term cultural-ecological inheritance that moderates $g$, is to change the minimum threshold at which a group's general intelligence reduces uncertainty associated with environmental complexity and how quickly increases in $g$ reduce the uncertainty associated with environmental complexity (see Fig 2a and 2b). In sum, the effect of $w_{div}$ is to reduce the uncertainty associated with environmental complexity, and hence the need for representational diversity to super-charge learning in a given Action Situation.

Eqs (1) and (2) formalize the propositions that (1) the higher the level of $g$ in a group the better individuals can search for and build functionally relevant representations of a system for decision making in an Action Situation. (2) As the complexity of an Action Situation increases, in terms of both social and environmental dimensions, groups need more cognitive representational diversity to find a solution to a governance challenge.

In the best of all worlds, groups would contain individuals with high $g$, *alpha* would be low, and would comprise all relevant stakeholders with the requisite $R_{div}$ needed to explore and supercharge learning. However, stakeholders, as noted above, also vary in their underlying goals and interests. This dimension of cognitive representational diversity can lead group members to focus on achieving different solutions or even engaging in deception, blocking the benefits of other dimensions of diversity by generating stalled positions, lack of communication, and mutual distrust of alternative solutions [16, 25, 39].

**Theory of mind.**   The distance between each individual stakeholder's goals (legitimate solutions), as noted above, are representational gaps [16, 24, 40]. The higher the representational diversity needed in a given action situation, the more likely the presence of representational gaps, and the more these gaps might be difficult to close. Eq 3 captures the exploration–exploitation tradeoff of this dynamic.

Eq 3 states that the representational gap in a given Action Situation is a function of the representational diversity ($R_{div}$) needed within the situation less the ability of agents to draw on their social intelligence ($ToM$) to form joint goals. Specifically, we assume that groups formed by individuals with higher $ToM$ better reduce representational gaps. Again, it is important to note that while $ToM$ plays a key role in the ability of groups to reduce representational gaps [13, 16], the effect of $ToM$ is assumed to be non-linear and moderated by group characteristics ($\beta$) that affect the ability of groups to reduce such representational gaps ($w_{gap}$) and harness the benefits of diversity ($R_{hdg}$). The parameter $\beta$, thus, describes the ability to generate a positive group culture as a result of past interactions between agents affecting, for example, levels of trust and shared narratives.

Eq 4 describes the effects of $ToM$ and $\beta$ on $w_{gap}$ (Fig 3a). The sigmoid relationship between $ToM$ and representational gaps captures the finding that a minimum level of ToM is required for groups to communicate effectively and form a joint goal [13]. In our model, the higher the level of $\beta$, the higher the level of distrust and the more fractious the narratives that individuals bring into an Action Situation. Thus, as $\beta$ increases it takes an ever higher level of minimum $ToM$ for a group to start reducing their representational gaps and harness the benefits of diversity (see Fig 3b). Mathematically,

$$R_{hdg} = R_{div} - \frac{ToM \times w_{gap}(ToM)}{\max(ToM)/2} \tag{3}$$

where $-20 \leq R_{hdg} \leq 20$, $ToM \geq 0$ and

$$w_{gap} = \max(ToM)\left[\tanh\left(\frac{ToM}{\max(ToM) - ToM}\right)\right]^{\beta} \tag{4}$$

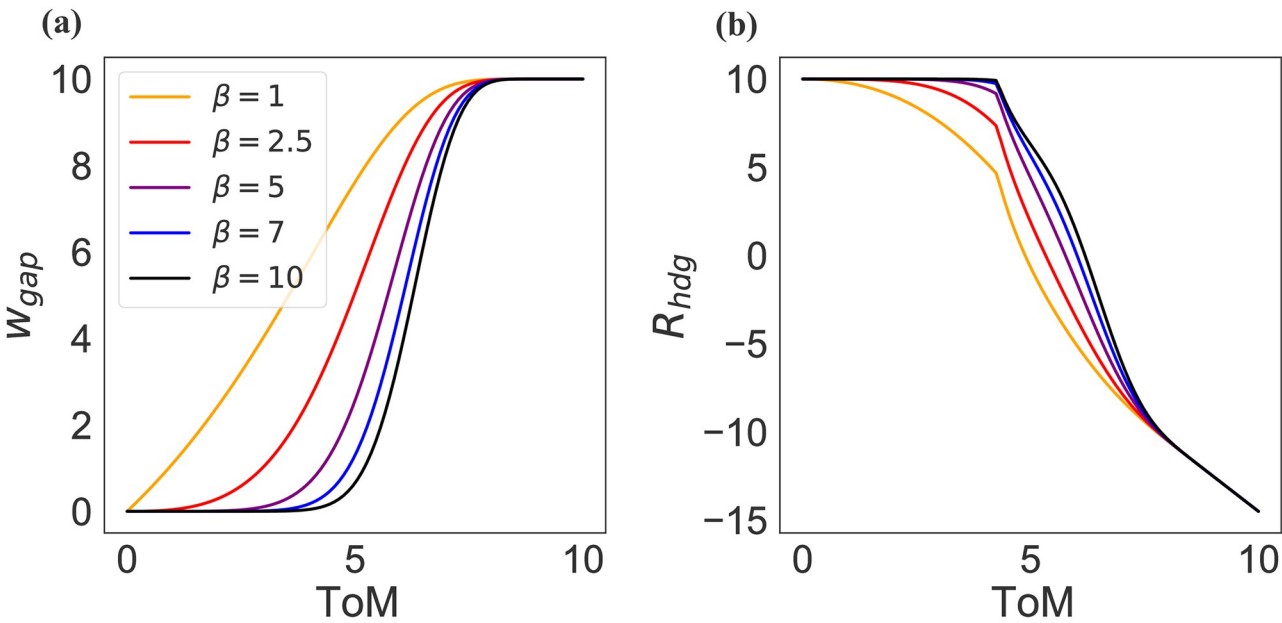

**Fig 3. (a) Effect of group characteristics $\beta$ and theory of mind ($ToM$) on the ability of groups to close representational gaps ($w_{gap}$) and (b) to harness the benefits of representational diversity ($R_{hdg}$).** The higher the level of $\beta$ the higher are distrust and fractured identities and the higher the minimum level of $ToM$ required to reduce representational gaps and harness the benefits of diversity in an action situation. Figure was generated keeping $E_{cpx} = S_{cpx} = 5$.

where $\beta \geq 0$. In short, Eqs (3) and (4) formalize the propositions that (1) the higher the level of representational diversity in an Action Situation, the higher the potential for representational gaps. (2) The higher the level of $ToM$ in a group, the better individuals form joint goals–moderated by $\beta$–and are able to harness the benefits of diversity to find solutions to governance challenges.

In sum, cognitive abilities such as general intelligence ($g$) and theory of mind ($ToM$) together with the ability of groups to search for functionally relevant and diverse representations of a problem ($\alpha$), as well as trust and shared narrative ($\beta$), allow groups to harness the benefits of higher cognitive representational diversity to solve problems of increasing complexity. The ability of groups to solve complex problems is, then, directly dependent upon the complexity of a problem faced in an Action Situation (both social and environmental complexity) and the ability of groups to close representational gaps in order to harness the benefits of the cognitive representational diversity needed to solve a problem of a given complexity. Given Eqs (1)–(4), we can write a group's average problem solving ability ($GPSA$) in relation to governance challenges within an Action Situation as a function of social and environmental complexity ($S_{cpx}$ and $E_{cpx}$) and cognitive abilities ($g$ and $ToM$) moderated by cognitive tools ($\alpha$ and $\beta$) as:

$$GPSA = 1 - \frac{(E_{cpx} + S_{cpx} + R_{hdg}(S_{cpx}, E_{cpx}, g, ToM) + 20}{60} \tag{5}$$

Where 20 and 60 are normalization parameters. The formalization presented in Eqs (1), (3) and 5, bridges and synthesizes work from organizational management, cognitive science, and collective action studies [5–9, 11, 13, 15, 16, 22, 25, 41]. If stakeholders share the same underlying goals, then groups with higher representational diversity experience an increase in

collective performance thanks to their ability to explore a wider "solution space" that increases the group's overall knowledge of a system [5, 6, 9].

## Model results

The above model illustrates the following insights (Fig 4). Averaging over all values of $\alpha$ and $\beta$: (1) Increased socio-environmental complexity requires increased representational diversity and, ultimately, higher cognitive abilities (both $g$ and $ToM$). Higher cognitive abilities allow for a better assessment of the problem, and also increase the ability of groups to act collectively and collaborate, both key components for managing complex systems [42]. It is worth mentioning here that $ToM$ has a bigger influence on group problem solving ability than $g$, especially when problems become more complex. (2) When complexity relates to the social system,

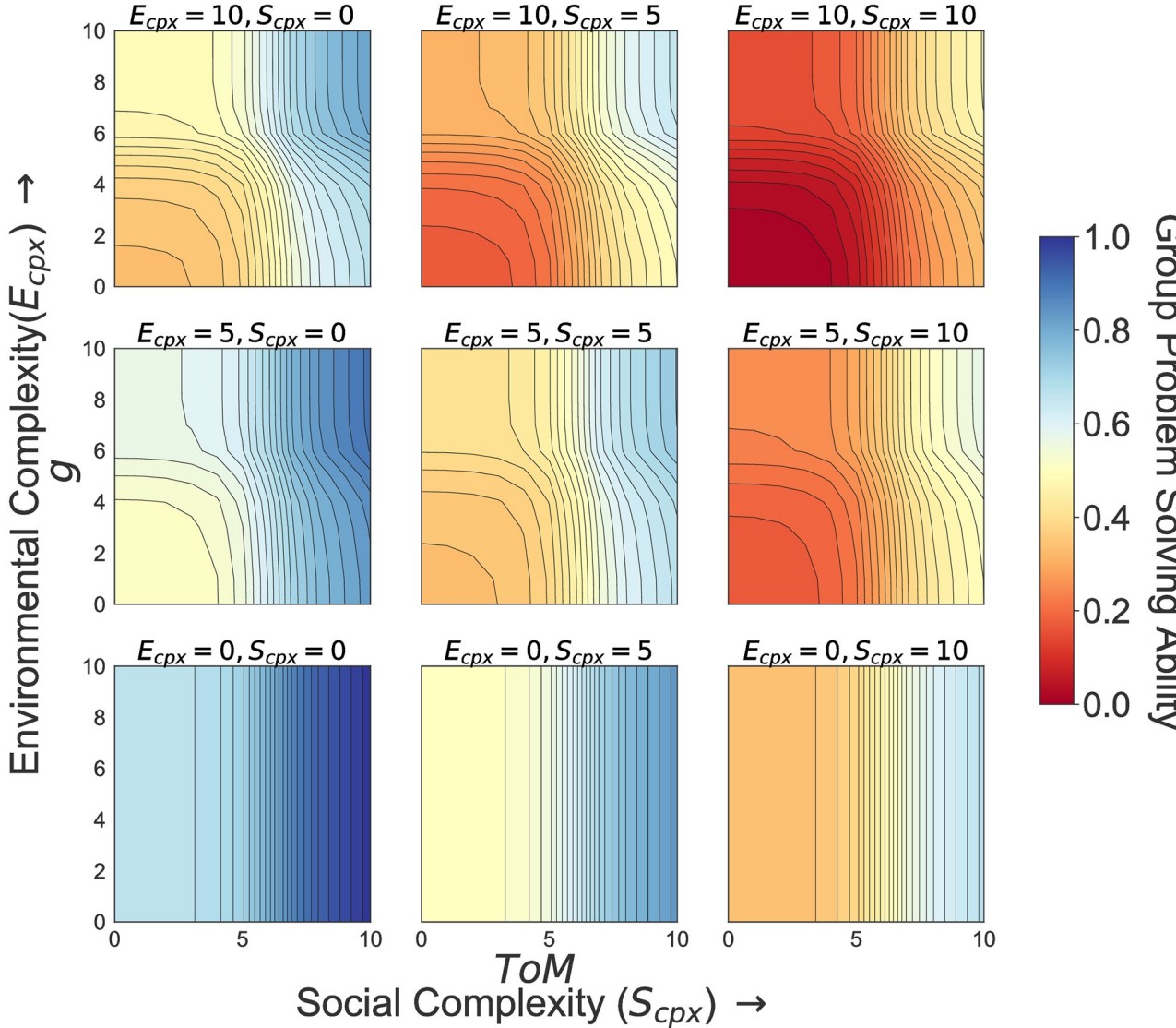

**Fig 4. Effect of $g$ and $ToM$ for different levels of social and environmental complexity.** $E_{cpx}$ = environmental complexity, and $S_{cpx}$ = social complexity averaged over $\alpha$ and $\beta$ assuming values [1, 10]. The results depicted in Figure. 4 are all based on the following parameter range: $g$, $ToM$ = [0, 10]; $\alpha$ and $\beta$ = [1, 10] for selected values of $E_{cpx}$ and $S_{cpx}$.

*ToM* drives a group's problem solving ability. This is especially relevant when the dynamics of an environmental problem are known and predictable (bottom row, Fig 4). When environmental complexity increases, even in simple social settings, increased levels of *g* and *ToM* are required for increasing group problem solving ability (first column, Fig 4).

(3) Holding social and environmental complexity equal, group problem solving ability is also affected by the ability to search for and build functionally relevant representations of a system $\alpha$, and by the ability of groups to create a positive group culture (i.e. levels of trust and shared narratives within an action situation), $\beta$. Fig 5 illustrates that lower levels of $\alpha$ and $\beta$ *increase* a group's ability to solve collective governance problems and reduce the need for higher levels of both *g* and *ToM* (see also S1 and S2 Figs in S1 File where $\alpha = \beta = 1$, S3. Fig in S1 File where $\alpha = \beta = 5$, and S4. Fig in S1 File where $\alpha = \beta = 10$). These modeled results highlight how cognitive abilities entangle with group context to affect collective governance. The modeled results are also consistent with controlled behavioral experiments that illustrate the positive impact of generalized trust in the first round of repeated games, but, thereafter, average performance depends upon behavioral responses, as groups draw on their cognitive abilities and repeated interactions, to learn and generate outcomes over time [11, 13, 43, 44].

Finally, (4) at high levels of social and environmental complexity group problem solving ability never reaches it maximum potential. High levels of cognitive abilities (*g* and *ToM*) and tools ($\alpha$ and $\beta$) do not guarantee reaching a solution, although higher levels of both aid towards it (Figs 4 and 5). Fig 5 illustrates that decreases in $\alpha$ and $\beta$ improve the effectiveness of cognitive abilities for solving a problem at a given level of complexity. Hence, the ability of groups to better monitor and assess the biophysical system as well as a positive group culture are all fundamental factors when it comes to solving the most complex problems faced in socio-environment systems. This empirical relationship captures our proposition that cognitive tools, whether technological adaptations that increase a group's ability to monitor, regulate, and assess biophysical system or institutional arrangements that increase trust and shared narratives, are fundamental for groups to support and enhance individual cognitive abilities (*g* and *ToM*) in a given Action Situation. In short, technology and institutions are the hardware upon which software (cognitive abilities) run. Therefore, there is a natural stop-gap on the ability of a software to solve a complex problem. This stop-gap can be (re)moved by upgrading/changing hardware. In turn, new hardware affects new software development.

The results of the model presented suggest testable hypotheses underlying the ability of groups to assess and find collective solutions to governance problems at different levels of complexity and at different scales. All else equal, we expect:

1. More complex problems require groups with higher cognitive abilities to collectively solve them. Hence, cognitive abilities become more salient the more complex problems become.

2. Cognitive representational diversity (i.e. the number of unique problem representations within and between individuals) underpins the ability of groups to assess and analyze complex environments and engage in collective action and find solutions to problems.

3. The ability of groups to understand and assess biophysical systems ($\alpha$), as well as generate a positive group culture ($\beta$) moderates the relationship between cognitive abilities and group problem solving ability in Action Situations.

## Model application to real world data

To partially evaluate the model developed, we fit Eq 5 to data at the small group, U.S. states, and country level. For this purpose we gather data related to *g* and *ToM* approximated by IQ

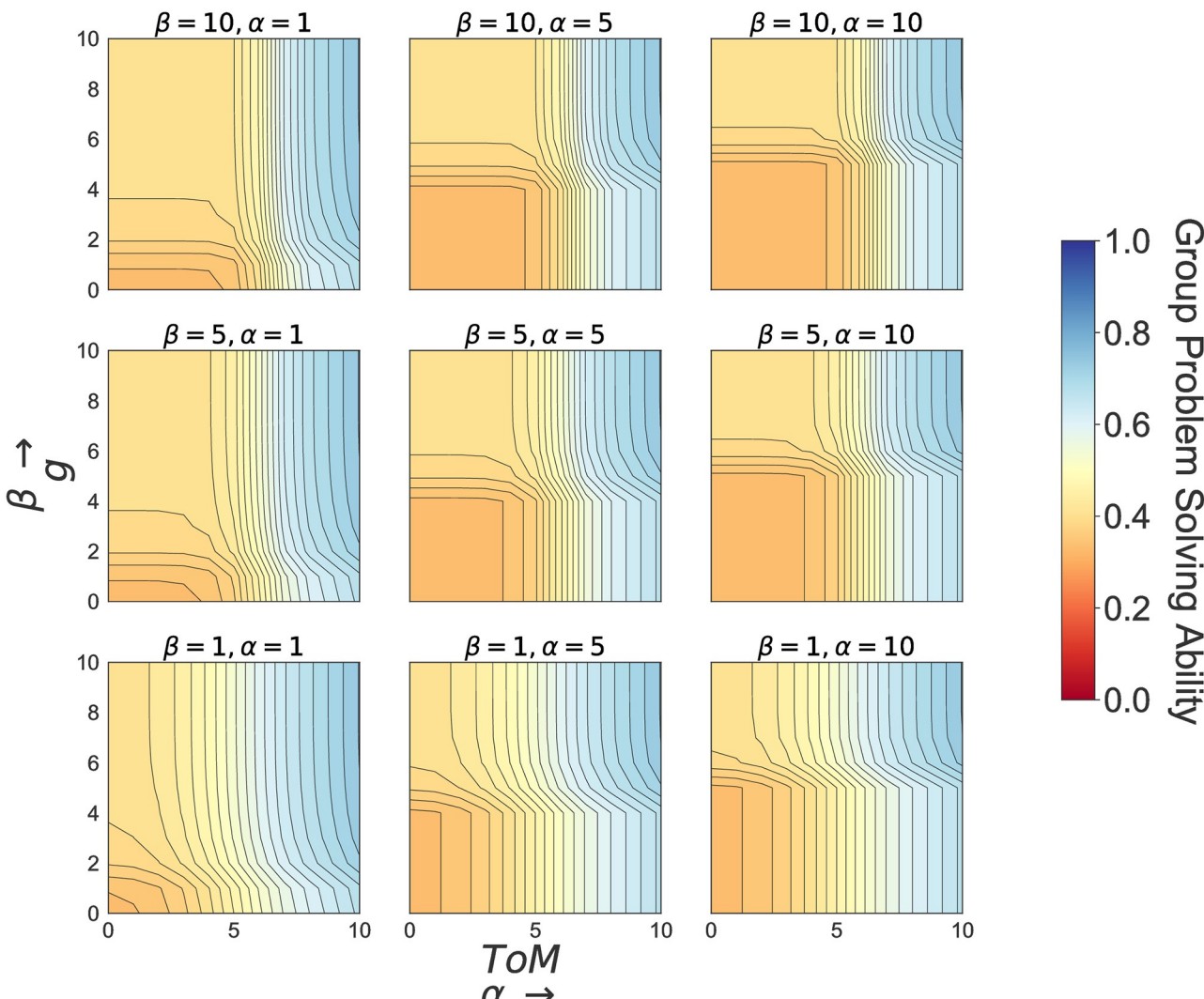

**Fig 5. Effect of *g* and *ToM* for different levels *α* and *β*.** The results depicted in Fig 5 are all based on the following parameter range: *g*, *ToM* = [1, 10], $E_{cpx} = S_{cpx} = 5$ for selected values of *α* and *β*.

and agreeableness metrics at the U.S. states [14, 45, 46] and country level [47, 48], and by ACT/SAT scores and an ad-hoc *ToM* test at the small-group level [11, 49–51]. Group problem solving ability is approximated by the ability to sustain and harvest resources at the small-group level based on data collected from behavioral experiments [11, 13]. At the U.S. state and country level, the problem solving ability of groups is approximated by their ability to effectively govern and provide public goods [52–54].

## Data

At the small-group level we estimate ToM via a short-story test [51], and we use ACT/SAT scores as an estimate of *g* as in [11, 13, 50]. We assess the ability of small groups to solve a governance problem by assessing how well they harvest and maintain density dependent resources. The ability to harvest resources at the small-group level in experimental settings can be the result of being "nice" or due to the presence of one person whom understands the game

and leads the other group members [55, 56]. However, we contend that being nice (or not being a "jerk"), for example, depends on *ToM*, and having a leader that understands the system reduces representational gaps that exist within the group.

At the U.S. state level we estimate *g* via standardized IQ tests [45], and we estimate *ToM* from the Big Five personality trait of agreeableness [46]. Here we measure *g* via scores based on averaged standardized tests. While this is not a direct measure of *g*, it is more general than specific IQ tests made to test specific aspects of IQ, and is highly related to IQ itself [14, 45, 57]. Agreeableness and *ToM* correlate more than any other Big Five personality trait, and it is the most appropriate estimate when no direct *ToM* measures are available [58, 59]. Agreeableness and *ToM* are associated neurologically [60], as similar brain regions are activated. Further, agreeableness is associated with the ability of individuals to process and assess social information and motivate more altruistic behaviors [59]. At the country level we use the same proxies (IQ and agreeableness) for *g* and *ToM*. Here *g* is derived from the national IQ database [47], and specifically we employ the mean between the IQ calculated from all samples weighted by data quality and number of individuals and the international school assessment studies. *ToM* is approximated by countries' averaged agreeableness [48]. Here we use an updated IQ measure version, that albeit still potentially incomplete, is the best available at the moment of writing. To further reduce issues with IQ and agreeableness measures at the country level we fit our model without data related to Sub-Saharan Africa as they are known to be the most affected by sampling and cultural biases [61–64].

Group problem solving ability is approximated by the ability to sustain and harvest resources at the small-group level based on data collected from behavioral experiments [11, 13]. At the U.S. state and country level, group problem solving ability is approximated by government effectiveness [52, 54]. Government effectiveness is a composite index assessing financial, infrastructure, and information management [52, 53], while at the country level government effectiveness is derived from expert perceptions related to public service quality, policy quality, implementation and credibility as well as public service independence from political pressures [54]. While there are multiple indicators of problem solving ability, we contend that government effectiveness is higher when a country/U.S. state is able to provide public goods and services on a regular basis in complex environments. Further, government effectiveness is highly related to other indices and metrics (e.g. rule of law, corruption indices, GDP) that relate to the ability of a group of individuals (here U.S. state or country) to solve complex problems. Thus, while we assess different outcome variables, namely managing and harvesting natural resources (small groups) and dealing with governance effectiveness (U.S. states and countries), both are related to the ability of groups to solve governance problems and hence, we contend, can be used to partially validate the model developed as they fundamentally approximate the same construct.

We re-scale the variables based on $\frac{x_i - x_{min}}{x_{max} - x_{min}}$, where $x_i$ is the value of observation $i$ and $x_{min}$ and $x_{max}$ are the minimum and maximum value of the vector $\vec{x}$. Finally, to conform to the model parameters and output, we then multiply *g* and *ToM* values by 10. Hence all data for *g* and *ToM* will assume values in the [0, 10] interval, while values for group problem solving ability (*GPSA*) will assume values in the [0, 1] interval. For distribution of *g*, *ToM*, and *GPSA* at the small group, U.S. states, and country level see S5 Fig in S1 File.

We fit the model to the data collected at the small group, state, and country level using lmfit and scikit-learn [65, 66]. We do this by employing a dual simulated annealing algorithm that elicits the parameter combination related to $S_{cpx}$, $E_{cpx}$, $\alpha$ and $\beta$ that best approximate the data. Simulated annealing is a numerical metaphor for a metallurgic process by which one increases or decreases the temperature of a metal to modify its structure and achieve a specific state (e.g.,

robustness level). The algorithm mimics this process by (1) starting with a random parameter configuration; (2) assessing neighbouring states of that configuration; (3) evaluating the fitness of the original and neighboring combination; and (4) choosing with a certain probability a configuration based on the difference between the fitness of each configuration [67–69]. The algorithm repeats the same process up to a pre-specified number of computation cycles. In order to fit the model to the data we devise a fitness function that assesses the difference between observed and predicted values for group problem solving ability *GPSA*. The fitness function is calculated as the average sum of squares between the estimated value stemming from the model depicted in Eqs (1), (3) and (5)–$\widehat{GPSA}$–and the actual value of *GPSA* (i.e. token harvested or government effectiveness) for each observation *i*: $Fit = \frac{\sum (\widehat{GPSA_i} - GPSA_i)^2}{N}$ where *N* is the number of observations in the real data. The dual annealing algorithm will then minimize *Fit*: *min*(*Fit*). Finally, we fit our model focusing on the behavioral response loop as presented in Fig 1, where *g* and *ToM* are considered variables, and social and environmental complexity, as well as $\alpha$ and $\beta$, are considered parameters (see Table 1).

## Implied assumptions and predictions

Importantly we assume that $S_{cpx}$ and $E_{cpx}$ as well as $\alpha$ and $\beta$ vary only across different levels of governance systems, and not within them. Second, we perform individual fit of Eq 5 to each small group, U.S. state and country separately and assess the variability of $S_{cpx}$, $E_{cpx}$, $\alpha$ and $\beta$ hence allowing us to assess the variability of complexity and cognitive tools within each governance level. Specifically we can then operationalize our hypotheses related to complexity levels ($E_{cpx}$, $S_{cpx}$) and cognitive tools ($\alpha$, $\beta$) to predict a specific set of outcomes based on the data analyzed:

1. Social and environmental complexity will be lower at the small group than at the U.S. state or country level: $S_{cpx,sg} < S_{cpx,ctr}$ and $E_{cpx,sg} < E_{cpx,ctr}$. Consequently:

   a. the need for representational diversity at the small group level will be lower than at the U.S. state or country level: $R_{div,sg} < R_{div,ctr}$.

   b. representational gaps at the small group level are less pronounced and present than at the U.S. state or country level, hence small groups are more able to harness the benefits of diversity: $R_{hdg,sg} > R_{hdg,ctr}$

2. Small groups will have a higher ability to develop functionally relevant representations of the system, and generate positive group culture than U.S. states or countries: $\alpha_{sg} < \alpha_{ctr}$ and $\beta_{sg} < \beta_{ctr}$.

**Table 1. Time-scale dependent model elements as parameters and variables.**

| Key Factors | Behavioral (fast loop) | Cultural and Ecological Inheritance (slow loops) | Definition |
|---|---|---|---|
| *GPSA* | Variable | Variable | Group ability to solve complex problems related to governance challenges |
| $E_{cpx}$ | Parameter | Variable | Environmental complexity |
| $S_{cpx}$ | Parameter | Variable | Social complexity |
| *g* | Variable | Variable | General intelligence |
| $\alpha$ | Parameter | Variable | Group ability to understand biophysical conditions (e.g. technology, education etc.) |
| *ToM* | Variable | Variable | Theory of mind |
| $\beta$ | Parameter | Variable | Group ability to generate a positive group culture (e.g. trust, shared identities etc.) |

3. Variability in social and environmental complexity is higher at the U.S state and country level with respect to the small group level: $Var(S_{cpx,sg}) < Var(S_{cpx,ctr})$ and $Var(E_{cpx,sg}) < Var(E_{cpx,ctr})$ given the larger range of governance challenges faced by U.S. states and countries compared to small groups.

4. Variability (variance) in $\alpha$ and $\beta$ is higher at the U.S. state and country level than at the small group level: $Var(\alpha_{sg}) < Var(\alpha_{ctr})$ and $Var(\beta_{sg}) < Var(\beta_{ctr})$

## Model fitting results

In summary, fitting the model to empirical data illustrates six main results (see also Tables 2 and 3). (1) our model fits the data reasonably well (being *Fit* between 0.037 and 0.063). (2) Social and environmental complexity are much higher and vary much more at the U.S. state and country level than with respect to small-groups. (3) Social complexity is basically absent at the small group level, but varies between different small groups when the model is fitted individually to each group. (4) The low $\alpha$ (= 2.3) at the small group level indicates that small groups are better equipped to use their general intelligence to build functionally relevant representations of the system compared to U.S. states ($\alpha$ = 3.2) and countries ($\alpha$ = 10). (5) $\alpha$ and $\beta$ vary more at the country level than at the small group level, indicating higher variability in conditions related to the ability of groups to build relevant representations of a governance system, as well as starting with a positive group culture. (6) These results showcase how higher levels of complexity require higher levels of representational diversity that can also generate representational gaps. These gaps can be closed, and groups can then work on increasing communication, trust, and sharing knowledge to further increase their ability to solve complex problems (in the model presented, this happens when $R_{hdg} < 0$). Based on these six results, we suggest that closing representational gaps in complex environments requires high levels of cognitive abilities (*g* and *ToM*), in tandem with the appropriate cognitive tools ($\alpha$ and $\beta$).

The six results listed above are mostly consistent with our predictions. For example, Table 2 illustrates that U.S. states and countries have higher social and environmental complexity than small group settings. Further, among small groups of undergraduate students, $\alpha$ is much lower (possibly indicating, on average, better education and capabilities to execute the small group task assigned in an experiment), while U.S. states and countries have higher levels of $\alpha$. In all cases, groups have identical $\beta$ values, suggesting very similar (or identical) levels of characteristics affecting the generation of positive group culture (see Table 2). While we would expect $\beta$ to actually decrease as group size decreases we do not observe this in the data analyzed here, countering the intuition that small groups have often higher positive group culture than larger groups.

**Table 2. Mean best parameter values after 1000 realization of model fitting via simulated annealing.**

|            | Small groups | U.S. states | Countries |
|------------|:------------:|:-----------:|:---------:|
| $E_{cpx}$  | 2.59412      | 10.00000    | 10.00000  |
| $S_{cpx}$  | 0.00000      | 1.60920     | 1.29554   |
| $\alpha$   | 2.29527      | 3.18766     | 10.00000  |
| $\beta$    | 1.00000      | 1.00000     | 1.00000   |
| *Fit*      | 0.03720      | 0.03789     | 0.06284   |

Note: parameters were allowed to vary as follows: $E_{cpx}$ = [0, 10], $S_{cpx}$ = [0, 10], $\alpha$ = [1, 10], $\beta$ = [1, 10].

Variance = 0.00000 for all parameter values over 1000 realization.

**Table 3. Variance of best parameter values for each small group, U.S. state, and country after 1000 realization via simulated annealing.**

|  | Small groups | U.S. states | countries |
|---|---|---|---|
| $Var(E_{cpx})$ | 4.464 | 9.983 | 15.203 |
| $Var(S_{cpx})$ | 4.294 | 7.516 | 13.659 |
| $Var(\alpha)$ | 0.494 | 0.851 | 1.910 |
| $Var(\beta)$ | 4.390 | 3.854 | 6.968 |

Note: parameters were allowed to vary as follows: $E_{cpx} = [0, 10]$, $S_{cpx} = [0, 10]$, $\alpha = [1, 10]$, $\beta = [1, 10]$. Except for $\beta$, the variance is lowest in small groups, followed by U.S. states, and countries.

Consistent with expectations, Table 3 illustrates that the variation in social and environmental complexity in small groups is much lower than the variation in these parameters between U.S. states or countries. This is expected as the governance challenges of U.S. states and countries vary widely compared to the challenge that small groups were faced with. Similarly, Table 3 illustrates that the variability of $\alpha$ between groups increases from small groups to U.S. states to countries. Finally, inconsistent with our predictions, small groups display less between group variation in $\beta$ than countries, but slightly more variation than U.S. states.

Consistent with predictions 1a and 1b, Fig 6 illustrates the need for higher representational diversity when groups face more complex governance challenges, such as the those challenges faced by U.S. states and countries. Complex governance challenges can also associate with larger representational gaps, and hence require higher levels of both $g$ and $ToM$ to be addressed.

The difference in $\alpha$ between small-groups, U.S. states, and countries affects how $g$ influences ecological complexity and the need for representational diversity ($R_{div}$). In small groups,

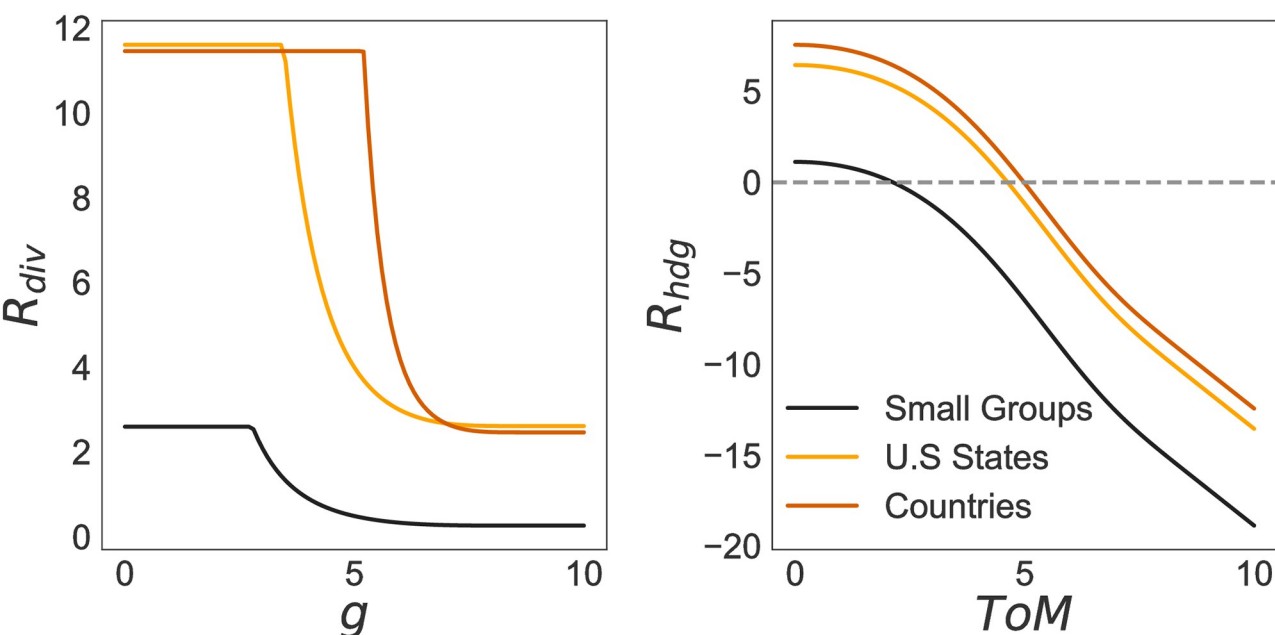

**Fig 6. Effect of $g$ on the need for $R_{div}$, and $ToM$ on the ability to close representational gaps $R_{hdg}$ in solving complex problems for the best-fit parameter values reported in Table 1.** The dotted line indicates where $R_{hdg}$ is not detrimental to the ability of finding solutions anymore (i.e. = 0). Additional $ToM$ is then spent to better achieve solutions rather than closing representational gaps.

increases of $g$ beyond 2.7, coupled with low levels of environmental and social complexity greatly reduce the need for representational diversity to search the potential solution space of a problem. Further, the effect of $g$ on representational diversity is different in small groups than in U.S. states and countries ($\alpha_{smlgrp} < \alpha_{U.S.} < \alpha_{ctr}$). This implies that small groups, and secondly U.S. states, are better equipped to build functionally relevant system representations to solve the governance challenges captured by the data on government effectiveness than countries.

The effect of $g$ differs not only because of the different levels of ecological and social complexity, but also because of how well equipped small groups, U.S. states, and countries are to build functionally relevant understandings of a governance system. In U.S. states and countries characterized by higher levels of environmental and social complexity, there is a minimum $g$ required before there is any beneficial effect of general intelligence on reducing the environmental complexity of the problem, and hence the need for representational diversity. Once that threshold is reached (U.S. states > 3.4 and countries > 5.1), $g$ sharply reduces the need for representational diversity via reducing $E_{cpx}$. However, once $g$ reaches a second critical value (> 6 for U.S. states and > 7 for countries) the reduction of $E_{cpx}$ is negligible for each increase in $g$. In other words, while small groups relying on their understanding of the environment are able to reduce $E_{cpx}$ and hence the need for representational diversity, this is not so for U.S. states and countries where $g$ is never sufficient to understand a system and the effect of $E_{cpx}$ on $R_{div}$. Higher complexity problems require representational diversity to increase groups' problem solving ability independent of the level of $g$.

The different levels of representational diversity needed to solve governance problems at the small group vs. U.S. states and countries also leads to differences in the relationship between *ToM* and the ability of groups to close representational gaps and harness the benefits of diversity. Specifically, at low levels of social and environmental complexity (i.e. in our examples, the small group level) low levels of *ToM* reduce representational gaps to 0, and *ToM* > 2 will increase the ability of a group to harness the benefits of diversity. On the other hand, when environmental and social complexity are higher (i.e. in our example at the U.S. state and country level) higher levels of *ToM* (approximately >5) are needed to close potential representational gaps and start harnessing the benefits of diversity.

## Conclusion

From biodiversity loss to climate change to racial inequalities, complex problems that require the collaboration and coordination of diverse stakeholders are ubiquitous in our modern world. These problems, while difficult to solve, are most importantly collective action problems: to be solved, they not only require technology, but the cooperation and coordination of actors with diverse value systems, beliefs, and objectives (i.e., high social complexity). Here, we presented a model that synthesizes relevant literature and generates testable hypotheses and avenues for future research on the interaction between cognitive abilities, diversity, and the social-ecological complexity of an action situation.

Our model suggests that larger groups need institutions and norms that serve as cognitive enhancers to tackle the environmental and social complexity they deal with. These "enhancers" are investments in institutional arrangements (e.g., Ostrom's Institutional Design Principles [42, 70, 71]), that reduce social complexity, as well as investment in public infrastructure such as schools, libraries, dams, canals and monitoring technology [72, 73] that reduce environmental complexity. These investments modify $\alpha$ and $\beta$, and the ability of groups to effectively draw on their cognitive abilities to solve governance challenges. In terms of the IAD we speculate that mechanisms of inter-group competition should lead to cultural and ecological

inheritance processes that affect the scale and effectiveness of such enhancers. Thus, institutions and technologies not only act upon a group's ability to search for and build functionally relevant representations of a system and positive group culture, but also moderate $g$ and $ToM$ at the group level. Further, our model suggests that representational diversity is always necessary to solve the most complex problems, thus groups constantly face a tension between supercharging learning and legitimacy through diverse representations, and representational gaps potentially leading to conflicts and mis-understandings generated by competing interests and goals.

Our results demonstrate a clear difference between problems at the small group level, where challenges are lower in levels of social and environmental complexity, and U.S. states and countries. While the results relating to small groups relate to groups of 4 to 8 individuals, other research has shown that these results may well apply to groups up to around 150 individuals [74, 75]. On the other hand, in large groups characterized by high social and environmental complexity, high levels of $g$ and $ToM$, while helpful, do not guarantee the ability to solve governance challenges especially at higher levels of social and environmental complexity. It is important to note that the model presented and its application is only a first step towards uncovering the relationship between cognition, technology, institutions and the ability of groups to solve governance problems in socio-environmental systems. Specifically, here we partially validate an instance of the behavioral response loop (see Fig 1), where cognitive tools and complexity are considered constant). We also assess $\alpha$ and $\beta$ as individual parameters while, they are in fact the result of multiple interacting factors. Future work should focus on better defining and unpacking the factors represented by cognitive tools, and assess the feedback and interaction between complexity, cognitive tools, and cognitive abilities over time, as well as the potential path dependencies that the cultural and ecological inheritance loops may generate, and how this may interact with the behavioral response loop (see Fig 1).

Expanding our model results to the inheritance loop of the IAD as represented in Fig 1, one could speculate that increased investment in institutional and public infrastructure can, in time, generate path dependencies and hard to change systems model representations leading to locked-in problem solving strategies. In fact, institutional arrangements and public infrastructure investments are the result of group interaction, hence, are the result of time-space specific Action Situations coupled with social and environmental complexity that can fail in the face of changing environments. Fig 1 implies that knowledge can be encoded in both institutions and public infrastructure evolved to reduce both social and environmental complexity. This, however, can lock-in systems and, in the long term, reduce the ability of groups to adapt to new problems and complexities (see also [76, 77]).

Ultimately, we highlight the key role of cognitive abilities (represented by $g$ and $ToM$) and the ability of groups to make use of those abilities ($\alpha$ and $\beta$) as building blocks towards solving complex governance challenges. Higher levels of cognitive abilities allow for increased representational diversity and the reduction of representational gaps, increasing the ability of groups to act collectively and solve complex governance challenges in socio-environment systems.

## Supporting information

**S1 File. Contains all the supporting figures and captions.**
(PDF)

**S1 Data.**
(ZIP)

## Author Contributions

**Conceptualization:** Jacopo A. Baggio, Jacob Freeman, Thomas R. Coyle.

**Formal analysis:** Jacopo A. Baggio, Jacob Freeman.

**Funding acquisition:** Jacopo A. Baggio, Jacob Freeman, Thomas R. Coyle.

**Methodology:** Jacopo A. Baggio, Jacob Freeman, John M. Anderies.

**Writing – original draft:** Jacopo A. Baggio, Jacob Freeman, Thomas R. Coyle, John M. Anderies.

**Writing – review & editing:** Jacopo A. Baggio, Jacob Freeman, Thomas R. Coyle, John M. Anderies.

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
