## [Decision Letter · Decision Letter 0]

2 Aug 2021

PONE-D-21-18465

Harnessing the benefits of diversity to address socio-environmental governance challenges

PLOS ONE

Dear Dr. Baggio,

Thank you for submitting your manuscript to PLOS ONE. After careful consideration, we feel that it has merit but does not fully meet PLOS ONE’s publication criteria as it currently stands. Therefore, we invite you to submit a revised version of the manuscript that addresses the points raised during the review process.

Both reviewers appreciate the potential of your contribution for publication but raise a few concerns, including the need to clarify the conceptual model behind your exercise, your assumptions, aspects related to the dependent variable, and which part of your conclusions are data-proof as opposed to conjectures/interpretations.

We look forward to receiving your revised manuscript.

Kind regards,

Sergio Villamayor-Tomas

Academic Editor

PLOS ONE

Journal Requirements:

"NO" 

 This information should be included in your cover letter; we will change the online submission form on your behalf

"JAB, JF and TC acknowledge support from the National Science Foundation Grant SMA-1620457"

"JAB, JF and TC acknowledge support from the National Science Foundation Grant SMA-1620457.

https://www.nsf.gov/awardsearch/showAward?AWD_ID=1620457&HistoricalAwards=false

4. We note that you have stated that you will provide repository information for your data at acceptance. Should your manuscript be accepted for publication, we will hold it until you provide the relevant accession numbers or DOIs necessary to access your data. If you wish to make changes to your Data Availability statement, please describe these changes in your cover letter and we will update your Data Availability statement to reflect the information you provide

5. The github data link is not working, provide working link.

Additional Editor Comments (if provided):

Both reviewers appreciate the potential of your contribution for publication but raise a few concerns, including the need to clarify the conceptual model behind your exercise, your assumptions, aspects related to the dependent variable, and which part of your conclusions are data-proof as opposed to conjectures/interpretations.

Reviewers' comments:

Reviewer's Responses to Questions

**Comments to the Author**

1. Is the manuscript technically sound, and do the data support the conclusions?

Reviewer #1: Yes

Reviewer #2: Partly

2. Has the statistical analysis been performed appropriately and rigorously? 

Reviewer #1: Yes

Reviewer #2: I Don't Know

3. Have the authors made all data underlying the findings in their manuscript fully available?

Reviewer #1: Yes

Reviewer #2: No

4. Is the manuscript presented in an intelligible fashion and written in standard English?

Reviewer #1: Yes

Reviewer #2: Yes

5. Review Comments to the Author

Reviewer #1: In this manuscript authors present a theoretical model to understand the role of cognitive abilities and diversity to solve complex governance problems. The model is very well presented and the results obtained very interesings. Please, find below my minor comments:

- Authors could include a table listing and describing all the parameters included in their model.

- Althought not needed, authors could include a diagram summarizing the main results obtained.

Introduction

- Better explain the IAD framework so people from all backgrounds can undertand it

- Give examples of the fast loop in the IAD framework

Results

“However, in all cases, groups seem to have similar levels of trust and a positive group

Culture”: Please better explain the reason behind this result. I would expect that Beta values will increase as the size of the group decreases (?).

Reviewer #2: Dear authors,

It is with great interest that I have reviewed the manuscript “Harnessing the benefits of diversity to address socio-environmental governance challenges”, submitted to PLOS One as a Research Article.

The paper describes an abstract mathematical model for studying how cognitive abilities, representational diversity and levels of theory of mind affect group decision-making for solving complex problems. The model is fitted to data at three different scales of problem-solving: small groups of students, U.S. states and countries. The estimated parameters vary across the cases in line with how the properties that they represent would have been expected to vary in real life. The results suggest that representational diversity plays an important role when socio-environmental complexity is high. At the other end, high cognitive abilities and theory of mind are not alone enough to enhance problem-solving.

The study addresses an important topic and can certainly contribute to advancing our understanding of the factors that enable group problem-solving for sustainability. In addition, I find that the results of the model fitting are able to partially validate the assumptions that the model is based on, the “first principles”, in the words of the authors.

However, I find that the manuscript still requires some work around the theoretical embedding of the concepts, as well as more reflexivity and clarity with respect to the constraints posed by the method when it comes to extrapolating conclusions to the real world.

Specifically, I would like to suggest that the authors give additional consideration to the following points:

1. Clarity of conceptual model

- While efforts have already been put into explaining the theoretical underpinnings (e.g. representational diversity and theory of mind) and the equations, as a reader I often get the feeling of shifting sands as concepts are used with multiple meanings interchangeably. For instance, in one place representational diversity seems to pertain to mental maps, in another to “predictive models” (lines 68-69), yet in another to different methodologies employed for making sense of reality (lines 64-66). In addition, some concepts are introduced unexpectedly, such as “collective intelligence”, or “ability to close representational gaps”, without clearly specifying how they interfere with the other variables in determining outcomes. It would be helpful to perhaps include a figure showing the conceptual model behind the equations: which variables are measured as outcomes, which are the independent variables, intervening variables etc. and what are the causal relationships between them. Some variables and parameters are already mapped on the IAD framework, but the latter doesn’t include the operationalization of the IAD elements (e.g. outcomes). It would be helpful to see not how alpha, beta, Ecpx etc. link to the IAD, but rather how the concepts behind them (general intelligence, access to schooling and technology, representational diversity, governance effectiveness) are assumed to be affecting each other.

2. Specification of the dependent variable

- It is not very clear what group problem solving ability represents in real-life. When fitting the model to real data, GPSA is approximated to the ability of a group to sustain resources (in the small group case) and to government effectiveness, respectively (in the country case). I understand the reasons for doing this and that no inter-comparable data is available. However, there is an important distinction between decision-making and outcomes/impacts. It would be useful to clarify ex-ante whether high GPSA refers to the results of the process or to the results in the environment – in IAD words, are the evaluative criteria applied to GPSA those related to the interactions branch or to the outcomes branch? I think this is particularly important because the outcomes will not depend solely on the decision/process in the action situation, but also on external factors (propensity to comply with policies, world events etc.).

3. More cautiousness in interpreting results

- The significance of various parameters in the model, e.g. alpha & beta, is a convention that we accept as readers. However, I think that it is difficult to later claim that the observed relationships between e.g. alpha and GPSA suggest a relationship between what alpha is assumed to represent and the effect. This is because alpha could have been any other factor that happens to have similar dynamics to the factors it is assumed it represents.

- Similarly, to what extent can we compare the values in Table 1 to each other across the three cases and draw conclusions about the implications of small or high values, if the outcomes component is mapped on different variables (ability to maintain a resource vs. government effectiveness)?

4. Better argumentation for the following foundational assumptions:

a. The more complexity increases, the more representational diversity is needed in order to have effective solutions. The argument behind this seems to be related to a process of eliciting knowledge about “unknown unknowns” (to an individual, but perhaps not to another) and avoiding blind spots with respect to the state of the system. However, I think this argument disregards the fact that the mapping of the problem space will be only as useful as is the knowledge pooled from multiple stakeholders valid and truthfully reported. In other words, it will not help the problem solving process if the diverse representations brought to the table are either completely invalid or intentionally misreported. Also, as you note as well, above a certain threshold, increased representational diversity might be counterproductive, as it may block the process due to conflicts.

b. Increased cognitive ability allows for a reduction of representational gaps. Here, as well, the question that comes to my mind is: is the skill/ability which is important here or is it rather the kind of knowledge possessed and its concordance with reality (as opposed to false beliefs)?

Disclaimer on a. and b.: it is possible that I am misreading/misinterpreting your foundational claims, but even then some clarifications would be helpful.

5. Data availability. Please note that the GitHub link provided is not working. I don't know whether you may have also included metadata for the cases with your submission, but I could only access the main manuscript body + the figures & the SI figures.

Details and minor points now follow, some of which add further granularity to the issues above. There are also a few typos in the manuscript that need to be corrected – I made a note of some of them, but I am sure I didn’t capture everything.

I hope you will find these comments useful in tightening up the manuscript and making it more accessible to a broader community of governance scholars.

Best wishes.

Detailed comments:

Introduction:

- Line 11: “the benefits and costs of diverse cognitive tools and abilities have been studied…” giving a brief overview of what these benefits and costs are would be helpful.

- Lines 13-14: “a formal model of how multiple dimensions of diverse cognitive tools and abilities interact […] is lacking”

o What are these dimensions that you are referring to? Could you give some examples already?

o Why is it important to develop a formal model, as opposed to researching these interactions by other methods?

- Line 15: typo “socio-envvironmental”

- Fig. 1 caption: g and ToM have not been introduced previously

- Lines 32-42: you might also want to refer to single/double-loop learning terminology/literature

- Line 36: typo “faculatively”

- Line 42: “water flow altered via irrigation” – someone might also argue that this is a fast process, depending on the time scale of the action situation, perhaps find a different example that is clearly slow?

- Lines 58-59: I think it would be good to dedicate a bit more attention here to defining g. It is not very clear at this point that you are referring to a property of individuals and not some emergent property of the group as a whole (although it becomes a bit clearer in the next lines).

- Lines 64-66: perhaps link to research on mental maps? See e.g. Doyle, J. K., & Ford, D. N. (1998). Mental models concepts for system dynamics research. System dynamics review: the journal of the System Dynamics Society, 14(1), 3-29. & derivative works.

- Lines 68-69: please indicate page number / source of the quote attributed to Hong and Page, I could not find it. It would be helpful to better explain what the claim refers to: what are “diverse predictive models” and how do these differ from the aforementioned “ways of representing the system”? I think that having correct knowledge about “what is” and making accurate predictions of “what might be” are two different things, and the latter does not depend solely on the accuracy of the first.

- Lines 71-73: is social complexity a measure of heterogeneity then?

- Lines 71-75: the claim that increased social complexity requires increased representational diversity due to reasons of fairness and legitimacy etc.: what about limits in communicating things to each other and getting consensus (if too much diversity). I think a distinction is needed: is representational diversity supposed to help the group perform better with respect to: a) understanding the system? OR b) collaboration? OR c) coordination?

- Line 80: the claim that diverse representations will necessarily reduce uncertainty associated with the environmental complexity: what if the representations are completely incorrect (unjustified beliefs) or they are intentionally communicated so as to distort facts – they might then have the opposite effect of reducing uncertainty, would you not agree? See also major point 4 above.

- Lines 89-90: exactly, diverse goals systems can “lead to stalled positions” – also: are representations independent of value systems or filtered by them?

- Line 96: social cognition has not been defined previously

- Line 106 and line 108: typo “socio-envvironmental”

The Formal Model

- Line 121: “need”: can there be a defendable need for representational diversity without discussing validity criteria (see my comment related to lines 68-69)

- Equation 2: it would be good to explicitly state in the text whether g is a set of gi of all group members or the average of gi of all group members, and also what max(g) represents in this context.

- Lines 134-148: it might be helpful to make a summary table with all the variables/parameters and what each represent (including alpha, beta, GPSA etc.), in addition to the figure suggested under major point 1.

- Line 164: here you introduce the concept of a “representational gap” for the first time and then it appears as an important element in the next subsection; perhaps discuss it briefly in the introduction – see also comment above on an additional figure to consider

- Line 168: given that “theory of mind” is such an important part of your formal model, it would be useful for the reader to have a bit more on this in the literature/introduction section.

Model Results

- Consider including more details about sensitivity analyses that you may have conducted. To what extent might the results be determined by your choices of those specific equations (e.g. non-linear you can model with many other equations) and specific parameter values (e.g. normalizing factors – 20 and 60 (why?) and how does the model behave across other areas of the parameter domain?

- Line 211: at what values are alpha and beta kept equal here (in comparison to Fig. S2, S3, S4)? Same question about Scpx and Ecpx from line 222 onward.

- Line 213: “(both g and ToM)”- also from the figure it appears that ToM has a bigger influence on the outcomes than g, perhaps mention that.

- Lines 218-219: “This is especially relevant when the dynamics of an environmental problem are known and predictable (first row, Fig.4).” – isn’t predictability associated with lower environmental complexity – should the reference have been to the bottom row in Fig.4?

- Lines 244-246: there seems to be a big jump here from the values of alpha and beta to the claim that we need technological and institutional advancements to enhance decision-making. This link, if kept, requires more argumentation.

- Line 254: typo “suggest”

- Lines 265-268: this is related to major point 3 above; I find it a bit problematic that many different factors are clustered into one simple parameter “alpha”, or “beta”, respectively. I can understand that such a decision could be made in order to keep the model simple when one wants to study other aspects, but to then try to prove a claim about the effect of these combined factors on group problem solving skills seems a bit too much. All that the model really shows is that alpha and beta affect GPSA, but alpha might have been defined as anything else (temperature in the room, for instance), it doesn’t mean that we proved something about how the represented property affects the outcome.

- Lines 295-297: I understand the need to use agreeableness as proxy variable for ToM, but then the interpretation of the results should be very cautious about saying anything about ToM as explanatory variable for the observed effects.

- Line 314: please explain how government effectiveness is assessed in this case /what indicators are used. Is it a property of governmental policies or their impact (outcome)? Outcomes might also depend on implementation (monitoring, enforcement) rather than policy design, in which case they may have little to do with g and ToM and the decision-making process.

- SI, Fig. 5: please add legend for the lines

- Line 333: typo “states”

- Line 334: typo “assesses”

Model Fitting Results

- Line 379: typo “to to”

- Line 394: “similar levels of trust and a positive group culture” – I mentioned this in an earlier comment: assigning this particular interpretation to beta was a convention, it could have been anything else; can we now draw a strong conclusion about the level of trust in the three groups observed in reality?

- Lines 396-398: Can we really compare the parameters given that outcomes are represented differently in the three data sets?

Conclusion

- Line 452: typo “hypotheses”.

- Line 476: “effective” problem solving – what is effective? what indicator is used for that?

- Lines 482-484: “increased investment in institutional and public…” – what in the model suggests this?

- Line 487: “constraining their ability to include novel system representations…” – what about social learning and information diffusion from individual outside the institution/group: strength of weak ties as bringing new information + social learning as shown in the slow loop of your IAD figure.

- Lines 493-494: “collective intelligence” - this also seems here to be yet another concept – perhaps also to add to a conceptual model of how all these variables interact

6. PLOS authors have the option to publish the peer review history of their article (what does this mean?). If published, this will include your full peer review and any attached files.

Reviewer #1: No

Reviewer #2: No

---

## [Author Response · Author response to Decision Letter 0]

19 Oct 2021

Note that for easy reading we have included a response to reviewer word document which the responses are in blu. Here are the same responses in text (harder to read, in my opinion). Also, please note that there may be small discrepancies (only related to spelling) between the tracked changes version and the non-tracked changes version of the manuscript

First we would really like to thank the reviewers and the editor for the constructive feedback. We are confident that the manuscript has been greatly improved thanks to their thoughtful suggestions and comments. 

Reviewer #1: In this manuscript authors present a theoretical model to understand the role of cognitive abilities and diversity to solve complex governance problems. The model is very well presented and the results obtained very interesting. Please, find below my minor comments:

- Authors could include a table listing and describing all the parameters included in their model.

REPLY: We add now a table summarizing the model parameters and variables depending on whether we assess the behavioral (fast) or inheritance (slow) loop. (table 1).

- Althought not needed, authors could include a diagram summarizing the main results obtained.

REPLY: We have not added a figure summarizing the main results obtained, as the main results pertain to Figure 4 and 5, and their partial validation is showcased in Figure 6.

Introduction

- Better explain the IAD framework so people from all backgrounds can undertand it

- Give examples of the fast loop in the IAD framework

REPLY: We have reconceptualized the IAD figure in order to explicitly relate it to complexity, cognitive tools and abilities. Further we add the following simple, clarifying example related to the behavioral response loop:

“The behavioral response loop refers to how groups adjust their actions to their given circumstances over days to a few years. Individuals interact, reach specific outcomes, evaluate, and respond. Community responses to epidemics can be thought of an example of the behavioral response loop. Individuals within some defined community, say a school district, change(or not) their behavior based on how they interpret the system (in this case how the epidemic is spreading) and their assessment of other’s intentions to cooperate (wear masks, get vaccinated, etc.)”

Results

“However, in all cases, groups seem to have similar levels of trust and a positive group

Culture”: Please better explain the reason behind this result. I would expect that Beta values will increase as the size of the group decreases (?).

REPLY: Indeed, this is an interesting result and we added text to highlight that this result counter the intuition that smaller groups are more likely to develop higher levels of positive group culture:

“In all cases, groups have identical β values, potentially indicating very similar (or identical) levels of characteristics affecting the generation of positive group culture (see table 2). While we would expect β to actually decrease as group size decreases we do not observe this in the data analyzed here, countering the intuition that small groups have often higher positive group culture than larger groups”

Reviewer #2: Dear authors,

It is with great interest that I have reviewed the manuscript “Harnessing the benefits of diversity to address socio-environmental governance challenges”, submitted to PLOS One as a Research Article.

The paper describes an abstract mathematical model for studying how cognitive abilities, representational diversity and levels of theory of mind affect group decision-making for solving complex problems. The model is fitted to data at three different scales of problem-solving: small groups of students, U.S. states and countries. The estimated parameters vary across the cases in line with how the properties that they represent would have been expected to vary in real life. The results suggest that representational diversity plays an important role when socio-environmental complexity is high. At the other end, high cognitive abilities and theory of mind are not alone enough to enhance problem-solving.

The study addresses an important topic and can certainly contribute to advancing our understanding of the factors that enable group problem-solving for sustainability. In addition, I find that the results of the model fitting are able to partially validate the assumptions that the model is based on, the “first principles”, in the words of the authors.

However, I find that the manuscript still requires some work around the theoretical embedding of the concepts, as well as more reflexivity and clarity with respect to the constraints posed by the method when it comes to extrapolating conclusions to the real world.

Specifically, I would like to suggest that the authors give additional consideration to the following points:

1. Clarity of conceptual model

- While efforts have already been put into explaining the theoretical underpinnings (e.g. representational diversity and theory of mind) and the equations, as a reader I often get the feeling of shifting sands as concepts are used with multiple meanings interchangeably. For instance, in one place representational diversity seems to pertain to mental maps, in another to “predictive models” (lines 68-69), yet in another to different methodologies employed for making sense of reality (lines 64-66). In addition, some concepts are introduced unexpectedly, such as “collective intelligence”, or “ability to close representational gaps”, without clearly specifying how they interfere with the other variables in determining outcomes. 

REPLY: The reviewer is right, we have now tightened the language throughout the manuscript (please see the tracked change version) in order to use terms consistently. Also, as pointed out by reviewer #1, we have better clarified the IAD and its correspondence with elements used specifically in this work, as well as making sure we define terms when introduced or better embed concepts in the text so as to avoid the “unexpected” introduction of such concepts. 

It would be helpful to perhaps include a figure showing the conceptual model behind the equations: which variables are measured as outcomes, which are the independent variables, intervening variables etc. and what are the causal relationships between them. Some variables and parameters are already mapped on the IAD framework, but the latter doesn’t include the operationalization of the IAD elements (e.g. outcomes). 

It would be helpful to see not how alpha, beta, Ecpx etc. link to the IAD, but rather how the concepts behind them (general intelligence, access to schooling and technology, representational diversity, governance effectiveness) are assumed to be affecting each other.

REPLY: This is a great suggestion and hence we have modified Fig 1which provides the conceptual model and details key variables (e.g., independent variables, intervening variables, outcomes).

Further, the revision clarifies not only relations between model parameters (e.g., alpha, beta) but also how they affect the concept behind them. (i.e. from the new Fig. 1 caption: “The IAD Framework (adapted from Ostrom) depicts how slow variables (i.e. biophysical conditions, attributes of the community, and rules in use) generate action situations which, in turn, generate interactions among agents for collective decision making and problem solving. In the adapted framework we specify environmental complexity $E_{cpx}$ as being a key attribute of biophysical conditions, social complexity $S_{cpx}$ as a key attribute of the community, and cognitive tools as a factor affecting rules in use. Cognitive tools refer to the mean traits of a group inherited as a consequence of past technological adaptations and social interactions. We define two sets of tool: (1) traits that can foster a better understanding of the biophysical environment and its dynamics ($\\alpha$), and traits that foster a positive group culture ($\\beta$). Complexity and cognitive tools result from cultural and social inheritance and constitute parameters that impact interactions between and within groups in specific action situations. Interactions in action situations depend on specific cognitive abilities (here the ability to correctly represent key dynamics within the environment ($g$), and the ability to model others intentions ($ToM$). Interactions give rise to specific outcomes that are partly ascribed to a group's problem solving ability ($GPSA$). Outcomes are then evaluated via specific criteria and potentially modified via learning and collaboration in the behavioral (fast) feedback loop

In the future we plan to perform an empirical study assessing how concepts behind alpha and beta are related to each other and actually affect alpha and beta directly. However, this falls beyond the scope of the current paper. 

2. Specification of the dependent variable

- It is not very clear what group problem solving ability represents in real-life. When fitting the model to real data, GPSA is approximated to the ability of a group to sustain resources (in the small group case) and to government effectiveness, respectively (in the country case). I understand the reasons for doing this and that no inter-comparable data is available. However, there is an important distinction between decision-making and outcomes/impacts. It would be useful to clarify ex-ante whether high GPSA refers to the results of the process or to the results in the environment – in IAD words, are the evaluative criteria applied to GPSA those related to the interactions branch or to the outcomes branch? I think this is particularly important because the outcomes will not depend solely on the decision/process in the action situation, but also on external factors (propensity to comply with policies, world events etc.).

REPLY: We now clarify GPSA as the outcome that depends indeed on both process and external outcomes. Here we only assess GPSA as a result of a process, and these process is then embedded in the wider environment that can affect it greatly. We now add this as a limitation of the model presented here in the conclusion as follows:

“It is important to note that the model presented and its application is only a first step towards uncovering the relationship between cognition, technology, institutions and the ability of groups to solve governance problems in socio-environmental systems. Specifically, here we partially validate an instance of the behavioral response loop (see Fig. 1), where cognitive tools and complexity are considered constant). We also assess α and β as individual parameters while, they are in fact the result of multiple interacting factors. Future work should focus on better defining and unpacking the factors represented by cognitive tools, and assess the feedback and interaction between complexity, cognitive tools, and cognitive abilities over time, as well as the potential path dependencies that the cultural and ecological inheritance loops may generate ,and how this may interact with the behavioral response loop (see Fig. 1)”

3. More cautiousness in interpreting results

- The significance of various parameters in the model, e.g. alpha & beta, is a convention that we accept as readers. However, I think that it is difficult to later claim that the observed relationships between e.g. alpha and GPSA suggest a relationship between what alpha is assumed to represent and the effect. This is because alpha could have been any other factor that happens to have similar dynamics to the factors it is assumed it represents.

- Similarly, to what extent can we compare the values in Table 1 to each other across the three cases and draw conclusions about the implications of small or high values, if the outcomes component is mapped on different variables (ability to maintain a resource vs. government effectiveness)?

REPLY: The reviewer is right, and now we are more cautions in how we interpret results. Also we wold like to point out that while we use different We would like to highlight that the partial validation is justified by ability of group to solve governance problems, as noted in the following text in the data section:

“Further, while we assess different outcome variables, namely managing and harvesting natural resources (Small Groups) and dealing with governance effectiveness (US States and Countries), both are related to the ability of groups to solve governance problems and hence, we contend, can be used to partially validate the model developed as they fundamentally "approximate" the same construct”

4. Better argumentation for the following foundational assumptions:

a. The more complexity increases, the more representational diversity is needed in order to have effective solutions. The argument behind this seems to be related to a process of eliciting knowledge about “unknown unknowns” (to an individual, but perhaps not to another) and avoiding blind spots with respect to the state of the system. However, I think this argument disregards the fact that the mapping of the problem space will be only as useful as is the knowledge pooled from multiple stakeholders valid and truthfully reported. In other words, it will not help the problem solving process if the diverse representations brought to the table are either completely invalid or intentionally misreported. Also, as you note as well, above a certain threshold, increased representational diversity might be counterproductive, as it may block the process due to conflicts.

b. Increased cognitive ability allows for a reduction of representational gaps. Here, as well, the question that comes to my mind is: is the skill/ability which is important here or is it rather the kind of knowledge possessed and its concordance with reality (as opposed to false beliefs)?

Disclaimer on a. and b.: it is possible that I am misreading/misinterpreting your foundational claims, but even then some clarifications would be helpful.

REPLY: We clarify the foundational assumptions throughout (see tracked changes). And highlight that we assume that individuals bring to the table “functionally relevant representation of the systems (i.e. system representations that are at least partially correct and that are not willingly misleading).” 

Further, reducing representational gaps is aided by ToM that encapsulate, moderated by beta, the ability to communicate and make sense of the different points of view and system representations brought at the table. 

5. Data availability. Please note that the GitHub link provided is not working. I don't know whether you may have also included metadata for the cases with your submission, but I could only access the main manuscript body + the figures & the SI figures.

REPLY: We are very sorry for this, the link should now work. Note that we submitted the tables and scripts used for the analysis during the prior submission. We have now changed permissions on the GitHub so that it should be accessible by reviewers. Also all files have been re-uploaded (the data file used for the partial validation as well as the python scripts used for the model and model fitting as well as generating the figure in the manuscript.

Details and minor points now follow, some of which add further granularity to the issues above. There are also a few typos in the manuscript that need to be corrected – I made a note of some of them, but I am sure I didn’t capture everything.

I hope you will find these comments useful in tightening up the manuscript and making it more accessible to a broader community of governance scholars.

Best wishes.

Detailed comments:

Introduction:

- Line 11: “the benefits and costs of diverse cognitive tools and abilities have been studied…” giving a brief overview of what these benefits and costs are would be helpful.

REPLY: We have modified the phrase in order to hint to a few examples of benefit and costs as follows: 

“Diverse stakeholder groups bring to the table different technologies and institutions (cognitive tools) and are composed by individuals with different levels and types of cognitive abilities (e.g., general and social intelligence) leading to, at times, divergent cognitive representations of a system. “While the benefits (e.g., increased system understanding, cooperation, ability to find novel solutions) and costs (e.g., conflict, stalled positions) of cognitive tools and abilities have been studied by psychologists, organizational scientists, economists, and sustainability”

- Lines 13-14: “a formal model of how multiple dimensions of diverse cognitive tools and abilities interact […] is lacking”

o What are these dimensions that you are referring to? Could you give some examples already?

REPLY: We eliminated the reference to dimensions, they would include what is incorporated in alpha and beta, but we think that is best to refer to these in more detail in the subsequent paragraphs rather than here. 

o Why is it important to develop a formal model, as opposed to researching these interactions by other methods?

REPLY: Formal models allow to structure theoretical arguments in a very precise way, so as to facilitate hypothesis testing that can then be done via other methods. They are not sufficient, alone, but integrated with case studies and primary data collection, behavioral experiments and qualitative understanding can shed light on the complexity of the problems we aim to solve / address.

- Line 15: typo “socio-envvironmental”

REPLY: Corrected

- Fig. 1 caption: g and ToM have not been introduced previously

REPLY: Fig 1 has been redone and caption completely rewritten.

- Lines 32-42: you might also want to refer to single/double-loop learning terminology/literature

REPLY: We could use terminology related to single and double loop learning as well as the literature, however, given the concepts and nature of the model here presented, we do not think that this would add to the present paper unless we reframe the IAD also within those terms. While this is definitely a worthy idea (a synthesis of other theoretical bodies and frameworks), we would prefer to leave it for future work. 

- Line 36: typo “faculatively”

REPLY: Corrected - now the loop has been renamed in Behavioral as it better reflects the model devised here.

- Line 42: “water flow altered via irrigation” – someone might also argue that this is a fast process, depending on the time scale of the action situation, perhaps find a different example that is clearly slow?

REPLY: Indeed, we have rephrased it as follows: 

“The cultural and ecological inheritance loop is defined by slower feedback processes in which behavioral adjustments affect the cultural dimensions of rules in use (e.g., institutions, accepted technologies) and attributes of a community (e.g., diversity of experiences, values, beliefs and objectives),as well as the biophysical dimensions of a system (e.g. changes in sea level rise or changes in snow pack)”

- Lines 58-59: I think it would be good to dedicate a bit more attention here to defining g. It is not very clear at this point that you are referring to a property of individuals and not some emergent property of the group as a whole (although it becomes a bit clearer in the next lines).

And 

- Lines 64-66: perhaps link to research on mental maps? See e.g. Doyle, J. K., & Ford, D. N. (1998). Mental models concepts for system dynamics research. System dynamics review: the journal of the System Dynamics Society, 14(1), 3-29. & derivative works.

REPLY: We have rephrased the paragraph and added the following text in order to better define cognitive tools and abilities here (also see track changes)

“These resources include what we call cognitive abilities, cognitive representational diversity, and cognitive tools developed over time in the social-ecological contexts that generate repeated, structurally similar Action Situations. Understanding the differences and relationships between these three concepts lies at the core of the model that we propose. 

Cognitive abilities refer to variation in the capacities of individuals to construct mental models of biophysical and social environments. Two of the most general abilities, discussed below, are general and social intelligence, theory of mind in particular. Cognitive representational diversity stems refers to the different ways that individuals describe and understand a system. For instance, two individuals may have a different understanding of water variability within an irrigation system, as well as how the system may be able to deal with such variability. One may see the system as robust to predicted changes and focuses on how such changes should be dealt with from a water withdrawal perspective. Another may assess the system as insufficient to deal with predicted changes in water availability and proposes to alter the physical infrastructure first. Finally, cognitive tools refer to the ability of groups to enhance cognitive abilities through technology, education, and institutions that help encode and monitor the dynamics of biophysical systems or foster a positive group culture, given a past history of trust and shared narratives. “ 

- Lines 68-69: please indicate page number / source of the quote attributed to Hong and Page, I could not find it. It would be helpful to better explain what the claim refers to: what are “diverse predictive models” and how do these differ from the aforementioned “ways of representing the system”? I think that having correct knowledge about “what is” and making accurate predictions of “what might be” are two different things, and the latter does not depend solely on the accuracy of the first.

REPLY: The reviewer is right, Hong and Page actually refer to diversity of perspectives and heuristic, but not diversity of predictive models. We have now corrected this imprecision as follows:

“In fact, groups formed by individuals that adopt different perspectives approach problems differently and, if they are able to communicate (see below) and share knowledge, they are, on average, more apt to find solutions than groups with similar levels of ability but lower diversity \\cite{Hong2004}”.

- Lines 71-73: is social complexity a measure of heterogeneity then? 

REPLY: We do assume that social complexity encompasses “multiple types of social heterogeneity”. However, if the reviewer intends “heterogeneity” to incorporate knowledge fragmentation, diversity of value systems, beliefs, and objectives, different ways to approach a problem and understand it then social complexity is a measure of heterogeneity. 

- Lines 71-75: the claim that increased social complexity requires increased representational diversity due to reasons of fairness and legitimacy etc.: what about limits in communicating things to each other and getting consensus (if too much diversity). I think a distinction is needed: is representational diversity supposed to help the group perform better with respect to: a) understanding the system? OR b) collaboration? OR c) coordination?

REPLY: We completely agree with this comment, in fact, while we state that representational diversity is needed to understand the system and potentially increase the solution space, and we also clarify that representational diversity is the basis for collaboration and coordination, however, we also state (in the following paragraph) that representational diversity is only half of the battle as diversity can bring issues when It comes to collaboration and/or coordination between diverse groups of actors. This is why we affirm that it is fundamental to introduce the concept of representational gaps, and how these are a fundamental feature of group problem solving ability, especially when it comes to potentially contested problems such as the ones related to socio-environmental systems. Representational gaps are also now introduced in the introduction and related to representational diversity as follows: 

“While diverse representations can supercharge learning, they can also be the result of divergent goals and/or value systems that lead to representational gaps. Representational gaps are defined as inconsistencies between how individuals perceive and assess the objectives of a system, furthering, in some cases, stalled positions and negatively affecting the possibility of forming joint goals and achieving solutions

(Weingart et al. 2008, Cronin and Weingart 2019).”

- Line 80: the claim that diverse representations will necessarily reduce uncertainty associated with the environmental complexity: what if the representations are completely incorrect (unjustified beliefs) or they are intentionally communicated so as to distort facts – they might then have the opposite effect of reducing uncertainty, would you not agree? See also major point 4 above.

REPLY: Yes, the reviewer is right, representation can be incorrect or intentionally misleading. This is why we clarify that representations should be functionally relevant, and now clarify that we assume this to mean that they are not completely incorrect nor are intentionally misleading. Further, as the reviewer notes below, it is important to highlight the role of representational gaps that need to be reduced or filled in order for the diversity of system representation to actually allow for reaching a “problem solution”.

- Lines 89-90: exactly, diverse goals systems can “lead to stalled positions” – also: are representations independent of value systems or filtered by them?

REPLY: In our opinion (and going back to the Stoics and Aristoteles) representations of a system are never independent from the value systems and beliefs or objectives of groups/individuals. That is why, while diverse representations are necessary, they can lead to difficult situatioins (the work of Cronin, Weingart and colleagues on organizations is quite enlightening from this point of view). 

- Line 96: social cognition has not been defined previously

REPLY: Given the main comment above, we have eliminated social-cognition (as well as collective intelligence right after), and paraphrased what we mean by it in order to increase the coherence of the overall paper. 

- Line 106 and line 108: typo “socio-envvironmental”

REPLY: Corrected

The Formal Model

- Line 121: “need”: can there be a defendable need for representational diversity without discussing validity criteria (see my comment related to lines 68-69)

REPLY: We have rewritten the sentence at lines 68-69 to clarify that the need is actually due to the fact that “ ..., groups formed by individuals who adopt different perspectives approach problems differently and, if they are able to communicate (see below) and share knowledge, they are, on average, more apt to find solutions than groups with similar levels of ability but lower diversity. We think that this clarifies the need for representational diversity without discussing validity criteria (for which we refer to previous literature, see cited work by Aminpour, Arlinghaus, and Page between others).

- 

Equation 2: it would be good to explicitly state in the text whether g is a set of gi of all group members or the average of gi of all group members, and also what max(g) represents in this context.

REPLY: We have clarified (before eq. 2) that g at the group level means average g of the individuals composing such group, and we clarify that max (g) is the maximum that value that g can reach “ g is the average group g and max(g) is the theoretical maximum value that group level g can reach”

- Lines 134-148: it might be helpful to make a summary table with all the variables/parameters and what each represent (including alpha, beta, GPSA etc.), in addition to the figure suggested under major point 1.

REPLY: Indeed, we add now a table summarizing the model parameters and variables depending on whether we assess the facultative (fast) or inheritance (slow) loop (table 1). 

- Line 164: here you introduce the concept of a “representational gap” for the first time and then it appears as an important element in the next subsection; perhaps discuss it briefly in the introduction – see also comment above on an additional figure to consider

REPLY: We thank the reviewer for pointing this out. We now introduce the term representational gap in the introduction, as these are the gaps that affect information sharing and processing and lead to stalled positions and less coherent decision making, specifically we introduce it in the solving social challenges section of the introduction:

“While diverse representations can supercharge learning, they can also be the result of divergent goals and/or value systems that lead to representational gaps. Representational gaps are defined as inconsistencies between how individuals perceive and assess the objectives of a system, furthering, in some cases, stalled positions and negatively affecting the possibility of forming joint goals and achieving solutions

(Weingart et al. 2008, Cronin and Weingart 2019).”

- Line 168: given that “theory of mind” is such an important part of your formal model, it would be useful for the reader to have a bit more on this in the literature/introduction section.

Indeed, we introduce Theory of Mind at the end of the introduction that we think, balances the different model aspects. We relate theory of mind to the ability of groups to solve small group tasks, display higher level of cooperation and manage common pool resources (see:

REPLY: “Finally, groups with higher levels of theory of mind display an increased ability to find collective solutions and solve tasks in small-groups [28–30], govern a common pool resource more effectively [11, 12], and display higher levels of cooperation across a range of contexts [31–34]. Theory of mind is the ability to couple with the mental states and anticipate the preferences of other actors in a system [35]. Highly developed theory of mind means richer models of the mental states of other actors, and is posed to reduce conflict, increase communication effectiveness, and reduce the costs of forming joint goals [11–13, 36, 37]”

Model Results

- Consider including more details about sensitivity analyses that you may have conducted. To what extent might the results be determined by your choices of those specific equations (e.g. non-linear you can model with many other equations) and specific parameter values (e.g. normalizing factors – 20 and 60 (why?) and how does the model behave across other areas of the parameter domain?

REPLY: Functional forms are loosely derived and hinted from our previous results (Baggio et al. 2019 and Freeman et al. 2020). Parameter values are bounded between 0 and 10, but can be bounded between 0 and 1 (does not change, actually given the nature of the tanh function). The normalization factors, they represent the min and max that can be achieved, hence if we change the max level of g and ToM than we would need to change the parameters. They are chosen to ensure that results are bounded between 0 and 1. Other areas of the parameter domain will only pertain to values of Scpx and Ecpx and they follow the trend presented in figure 4.

- Line 211: at what values are alpha and beta kept equal here (in comparison to Fig. S2, S3, S4)? Same question about Scpx and Ecpx from line 222 onward.

REPLY: We are sorry we were not clear enough. What we meant in line 211 is that the results portrayed in Figure 4 represent model outcomes averaged over all values of alpha and beta (we rephrased it as follows: “Averaging over all values of α and β: …”)

For the supplementary figures we clarify the values of alpha and beta in the caption of each figure. We have now added the information in the main text:

“see also Fig. S1, and Fig. S2 where α=β= 1, Fig. S3 where α=β= 5, and Fig. S4 where α=β= 10”

- Line 213: “(both g and ToM)”- also from the figure it appears that ToM has a bigger influence on the outcomes than g, perhaps mention that.

REPLY: Thank you for pointing this out, we now mention this when writing about figure 4: 

“ It is worth mentioning here that ToM has a bigger influence on group problem solving ability than g, especially when problems become more complex”

- Lines 218-219: “This is especially relevant when the dynamics of an environmental problem are known and predictable (first row, Fig.4).” – isn’t predictability associated with lower environmental complexity – should the reference have been to the bottom row in Fig.4?

REPLY: Yes, sorry, for some reason I thought first row being the bottom one. Corrected.

- Lines 244-246: there seems to be a big jump here from the values of alpha and beta to the claim that we need technological and institutional advancements to enhance decision-making. This link, if kept, requires more argumentation.

REPLY: Indeed, we rephrased it as follows to highlight the supporting role of technology and institution: “Hence, the ability of groups to better monitor and assess the biophysical system as well as a positive group culture are all fundamental factors when it comes to solving the most complex problems faced in socio-environment systems. This empirical relationship captures our proposition that cognitive tools, whether technological adaptations that increase a group’s ability to monitor, regulate, and assess biophysical system or institutional arrangements that increase trust and shared narratives, are fundamental for groups to support and enhance individual cognitive abilities ( g and ToM) in a given Action Situation”.

- Line 254: typo “suggest”

REPLY: Changed

- Lines 265-268: this is related to major point 3 above; I find it a bit problematic that many different factors are clustered into one simple parameter “alpha”, or “beta”, respectively. I can understand that such a decision could be made in order to keep the model simple when one wants to study other aspects, but to then try to prove a claim about the effect of these combined factors on group problem solving skills seems a bit too much. All that the model really shows is that alpha and beta affect GPSA, but alpha might have been defined as anything else (temperature in the room, for instance), it doesn’t mean that we proved something about how the represented property affects the outcome.

REPLY: Indeed the reviewer is correct, to increase clarity we explicitly refer to only 4 factors that are exemplar for alpha and beta. Technology and education are represented by alpha, while level of trust and degree of shared narratives are folded into beta. This allows for more coherence and clarity of the overall arguments made here. Also, we would like to reiterate that we do not prove, but generate hypothesis that can then be tested. This is one of the advantage of models, that they allow for clear hypothesis to be made. In this case, one could hypothesize different factors contributing to alpha and beta and test which one are the most relevant (this not being the focus of this paper though). . 

- Lines 295-297: I understand the need to use agreeableness as proxy variable for ToM, but then the interpretation of the results should be very cautious about saying anything about ToM as explanatory variable for the observed effects.

REPLY: Indeed the reviewer is right, although agreeableness has been found related to ToM, it is not the same and it is based on self-assessment rather than specific ad-hoc tests. Hence we clarified the 6th point related to table 2 and changed slightly the phrasing so as to better clarify the nature of the speculation (i.e. cognitive abilities required are a consequence of the increased need for representational diversity and higher representational gaps). (see also the explanation of how agreeableness and ToM are related in text and from previous work as here mentioned (Freeman et al. 2016, p49)

1. The statements used to construct the agreeableness personality dimension on self-report surveys are about how well an individual believes they attend to others' mental states (Nettle & Liddle, 2008). For example, typical items on personality surveys are: “I sympathize with others' feelings;” “I feel others' emotions;” and “I am interested in people.”

2. At least three measures of social-cognitive ToM correlate at a moderate to high effect size with measures of agreeableness (Ferguson & Austin, 2010; Nettle & Liddle, 2008; Nettle, 2007). For example, Nettle and Liddle (2008) find that agreeableness correlates at 0.48 with direct measures social cognitive ToM, while other Big Five traits (openness, extroversion, conscientiousness) do not correlate with ToM. Ferguson and Austin (2010, Table 1) replicate this finding, observing significant correlations between a direct measure of social cognitive ToM and agreeableness, but weak and non-significant correlations between verbal ability (a factor of g) and ToM, as well as ability measures of emotional intelligence and ToM.

3. DeYoung et al. (2010) illustrate a neurological basis for arguing that agreeableness measures social-cognitive ToM. They document significant associations between agreeableness and the volume of regions of the brain involved in social cognition, such as the superior temporal sulcus and posterior cingulate cortex. They conclude that “these associations are consistent with the hypothesis that Agreeableness is associated with the social information processing that enables and motivates altruistic behavior” (DeYoung et al., 2010, 8).

Reference: Freeman, J., Coyle, T. R., & Baggio, J. A. (2016). The functional intelligences proposition. Personality and Individual Differences, 99, 46–55. https://doi.org/10.1016/j.paid.2016.04.057

- Line 314: please explain how government effectiveness is assessed in this case /what indicators are used. Is it a property of governmental policies or their impact (outcome)? Outcomes might also depend on implementation (monitoring, enforcement) rather than policy design, in which case they may have little to do with g and ToM and the decision-making process.

REPLY: Indeed, outcomes definitely can depend on monitoring, enforcement and the wider socio-political and economic environment. For the purpose of this work, we define government effectiveness following King et al. 2004 for the US States and Kaufmann et al. 2011 for Countries. 

“At the U.S. state and country level, group problem solving ability is approximated by government effectiveness (Knack and Keefer 2002, Kaufmann 2011). Government effectiveness is a composite index assessing financial, infrastructure, and information management \\cite{King2004, Knack2002}, while at the country level government effectiveness is derived from expert perceptions related to public service quality, policy quality, implementation and credibility as well as public service independence from political pressures \\cite{Kaufmann2011}.}.” 

- SI, Fig. 5: please add legend for the lines

REPLY: Added, sorry for missing this.

- Line 333: typo “states”

REPLY: Now states and countries are consistently lower case.

- Line 334: typo “assesses”

REPLY: Unsure whether this is a typo as it is the third person form

Model Fitting Results

- Line 379: typo “to to”

REPLY: Corrected

- Line 394: “similar levels of trust and a positive group culture” – I mentioned this in an earlier comment: assigning this particular interpretation to beta was a convention, it could have been anything else; can we now draw a strong conclusion about the level of trust in the three groups observed in reality?

REPLY: Indeed beta is a convention and rather than assessing similar levels of trust and positive group culture we now state the following:

Further, among small groups of undergraduate students,α is much lower (possibly indicating, on average, better education and capabilities to execute the small group task assigned in an experiment), while U.S. state sand countries have higher levels of α. In all cases, groups have identical β values, suggesting very similar (or identical) levels of characteristics affecting the generation of positive group culture (see table 2). While we would expect β to actually 

decrease as group size decreases we do not observe this in the data analyzed here, countering the intuition that small groups have often higher positive group culture than larger groups.

- Lines 396-398: Can we really compare the parameters given that outcomes are represented differently in the three data sets?

REPLY: While not perfect, in the data section we argue that 

“…while we assess different outcome variables, namely managing and harvesting natural resources (small groups) and dealing with governance effectiveness (U.S. states and countries), both are related to the ability of groups to solve governance problems and hence, we contend, can be used to partially validate the model developed as they fundamentally “approximate” the same construct.”

Conclusion

- Line 452: typo “hypotheses”.

REPLY: I am unsure if this is a typo as it is the plural of hypothesis.

- Line 476: “effective” problem solving – what is effective? what indicator is used for that?

REPLY: Indeed the reviewer is right, we do not define effective and it would be problematic to do so, hence we changed the phrasing to refer to our actual outcome variable: group ability to solve governance problems. 

- Lines 482-484: “increased investment in institutional and public…” – what in the model suggests this?

And 

- Line 487: “constraining their ability to include novel system representations…” – what about social learning and information diffusion from individual outside the institution/group: strength of weak ties as bringing new information + social learning as shown in the slow loop of your IAD figure.

REPLY: Given that we do not explicitly talk about social learning and connection density (on which social-learning can develop) nor we talk about weak and strong ties, we rephrased this part to be more coherent with our IAD conceptualization and highlight that these are not conclusions directly related to our model but based on extending the IAD conceptualization proposed to the slow feedback loop (Cultural and ecological inheritance):

“Expanding our model results to the inheritance loop of the IAD as represented in Fig. 1, one could speculate that increased investment in institutional and public infrastructure can, in time, generate path dependencies and hard to change specific systems model representations leading to locked-in problem solving strategies. In fact, institutional arrangements and public infrastructure investments are the result of group interaction, hence, are the result of time-space specific problems coupled with social and environmental complexity that can fail in the face of changing environments.

- Lines 493-494: “collective intelligence” - this also seems here to be yet another concept – perhaps also to add to a conceptual model of how all these variables interact

REPLY: We now avoid the concept altogether.

---

## [Decision Letter · Decision Letter 1]

14 Dec 2021

PONE-D-21-18465R1Harnessing the benefits of diversity to address socio-environmental governance challengesPLOS ONE

Dear Dr. Baggio,

Thank you for submitting your manuscript to PLOS ONE. After careful consideration, we feel that it has merit but does not fully meet PLOS ONE’s publication criteria as it currently stands. Therefore, we invite you to submit a revised version of the manuscript that addresses the points raised during the review process.

Dear author, please take this decision as an acceptance. I just wanted to be sure that you have an opportunity to amend the manuscript following the minor comments of Reviewer 2.I will not send the manuscript for revision again and assess your revised manuscript myself.

We look forward to receiving your revised manuscript.

Kind regards,

Sergio Villamayor-Tomas

Academic Editor

PLOS ONE

Journal Requirements:

Reviewers' comments:

Reviewer's Responses to Questions

**Comments to the Author**

1. If the authors have adequately addressed your comments raised in a previous round of review and you feel that this manuscript is now acceptable for publication, you may indicate that here to bypass the “Comments to the Author” section, enter your conflict of interest statement in the “Confidential to Editor” section, and submit your "Accept" recommendation.

Reviewer #1: All comments have been addressed

Reviewer #2: (No Response)

2. Is the manuscript technically sound, and do the data support the conclusions?

Reviewer #1: Yes

Reviewer #2: Yes

3. Has the statistical analysis been performed appropriately and rigorously? 

Reviewer #1: Yes

Reviewer #2: N/A

4. Have the authors made all data underlying the findings in their manuscript fully available?

Reviewer #1: Yes

Reviewer #2: Yes

5. Is the manuscript presented in an intelligible fashion and written in standard English?

Reviewer #1: Yes

Reviewer #2: Yes

6. Review Comments to the Author

Reviewer #1: (No Response)

Reviewer #2: Thank you very much for the revised manuscript. The authors did a great job at addressing previous comments and I find that the manuscript is much improved as a consequence. In particular, the introduction and theoretical sections read now much better and the revised Fig. 1 makes the argumentation easier to follow. I also appreciate the clearer definitions of alpha and beta and their conceptual significance. This time I was also able to access the data files and the model – thank you!

From my perspective, the manuscript is getting very close to publication. Below are a couple of minor points that remain to be addressed (1-2) and some additional suggestions (3-6).

1) Rdiv is such an important intervening variable that it has a dedicated section in the model description. On page 3, “cognitive representational diversity” is also listed as one of three important concepts, together with cognitive tools and cognitive abilities. Despite this, it is now difficult to relate Rdiv to the other concepts and the relationship to Scpx is also not very clear. For instance, is “increasing the stakeholder diversity” (p. 4) about increasing Scpx or about Rdiv, or about both but in different contexts (entire community vs. specific action situation)? Perhaps Rdiv might be added to Fig. 1 somewhere (e.g. in the “Action Situation” box?) and a few more conceptual clarifications can be made in text.

2) On page 5 it is stated: “greater socio-environmental complexity requires that groups increase their representational diversity to supercharge learning…” – and this is used as an assumption in the model. Later, a similar statement appears as a result (result 6 on p.14). A clarification or a refutation of circularity of argument is needed.

3) Page 6, where alpha is introduced: “The parameter alpha represents the cognitive tools affecting…” – perhaps good to mention already here that high value is actually “high restriction on access to tools”, because based on the definition one would initially assume that high value = more cognitive tools = better.

4) Consider checking Fig 4, Fig 5 and those in the supplementary information for readability in black/white.

5) Consider adding Rdiv, Rhdg, wdiv and wgap to Table 1.

6) A few typos and small errors remain, e.g. my previous comment about “hypotheses” – the singular form is now used instead of plural (p.17).

I think these are quick revisions to make, hence I look forward to seeing the manuscript published very soon.

Best wishes

7. PLOS authors have the option to publish the peer review history of their article (what does this mean?). If published, this will include your full peer review and any attached files.

Reviewer #1: No

Reviewer #2: No

---

## [Author Response · Author response to Decision Letter 1]

21 Dec 2021

Once again, we would really like to thank reviewers and the editor for the suggestions and comments that have really strengthen this manuscript. Please see our responses to the remaining queries below.

Reviewer #2: 

1) Rdiv is such an important intervening variable that it has a dedicated section in the model description. On page 3, “cognitive representational diversity” is also listed as one of three important concepts, together with cognitive tools and cognitive abilities. Despite this, it is now difficult to relate Rdiv to the other concepts and the relationship to Scpx is also not very clear. For instance, is “increasing the stakeholder diversity” (p. 4) about increasing Scpx or about Rdiv, or about both but in different contexts (entire community vs. specific action situation)? Perhaps Rdiv might be added to Fig. 1 somewhere (e.g. in the “Action Situation” box?) and a few more conceptual clarifications can be made in text.

Response: Thank you for this comment. We were not as clear as we could have been. Cognitive representational diversity actually has two dimensions that we have now clearly parsed. The first is mechanical--the different ways that individuals represent the mechanics of a system--and the second is values based--the different values and objectives that individuals bring to an action situation. We have now made this clear by revising the introduction, the text on page 3 where we define cognitive representational diversity, and in the text on page 4. It should now be clear that as social complexity increases, both dimensions of cognitive representational diversity increase. So, Scpx is external to the Action Situation. As scpx goes up, so does Rdiv. Rdiv includes both the mechanical and the value dimensions of cognitive representational diversity; this is why Rdiv is an important term in both equations 1 and 3. As Rdiv increases, mechanical and value representations of a system also increase; that is, these dimensions positively covary. This is why we argue that increasing Rdiv supercharges learning, as long as a process is in place for all to share their views, but increases in Rdiv may lead to conflict as values and objective clash even in cases where all can share their views. In essence ToM and $\\beta$ moderate the positive correlation of mechanical and value cognitive representational diversity (equations 3 and 4). More ToM and lower $\\beta$, in effect, weaken the correlation between mechanical cognitive div. and value/objective cognitive div. We have added representational diversity and gaps within the action situation to clarify further the argument made here. 

2) On page 5 it is stated: “greater socio-environmental complexity requires that groups increase their representational diversity to supercharge learning…” – and this is used as an assumption in the model. Later, a similar statement appears as a result (result 6 on p.14). A clarification or a refutation of circularity of argument is needed.

Response: NOTE: We don't present ``results on page 6, so I am assuming the reviewer means the ``model results" section.

We have now clarified that this is an assumption. On page 5 we present insights of visually representing the interaction between the formal model's variables and parameters. The `results' here are really modeled results or clarifications of our hypotheses based on how our assumptions lead to outcomes in the formal model space. We have attempted to make this clear in the model results section. 

3) Page 6, where alpha is introduced: “The parameter alpha represents the cognitive tools affecting…” – perhaps good to mention already here that high value is actually “high restriction on access to tools”, because based on the definition one would initially assume that high value = more cognitive tools = better.

Response: The reviewer is correct, we have amended the text. We have added the same language when we introduce the parameter beta.

4) Consider checking Fig 4, Fig 5 and those in the supplementary information for readability in black/white.

Response: Indeed, the only way to do this would be to have a monochromatic scale, given that Plos one is an online only journal, we are unsure whether this would allow for a clearer interpretation of the figures compared to the coolwarm scale chosen.

5) Consider adding Rdiv, Rhdg, wdiv and wgap to Table 1.

Response: This could be a great idea, however, table 1 represent parameters and variables of the model, while the four aforementioned model elements are a function of these parameters in the work presented, hence, it may create confusion to add them in the table, as they would entail rewriting the functions in the behavioral and cultural/ecological inheritance loops. 

6) A few typos and small errors remain, e.g. my previous comment about “hypotheses” – the singular form is now used instead of plural (p.17).

Response: Corrected, we hope to have addressed all remaining typos.

---

## [Editor Report · Decision Letter 2]

19 Jan 2022

Harnessing the benefits of diversity to address socio-environmental governance challenges

PONE-D-21-18465R2

Dear Dr. Baggio,

We’re pleased to inform you that your manuscript has been judged scientifically suitable for publication and will be formally accepted for publication once it meets all outstanding technical requirements.

Kind regards,

Sergio Villamayor-Tomas

Academic Editor

PLOS ONE

---

## [Editor Report · Acceptance letter]

26 Jan 2022

PONE-D-21-18465R2 

Harnessing the benefits of diversity to address socio-environmental governance challenges 

Dear Dr. Baggio:

I'm pleased to inform you that your manuscript has been deemed suitable for publication in PLOS ONE. Congratulations! Your manuscript is now with our production department. 

Kind regards, 

on behalf of

Dr. Sergio Villamayor-Tomas 

Academic Editor

PLOS ONE